# Unexplained hydrogen isotope offsets complicate the identification and quantification of tree water sources in a riparian forest

Adrià Barbeta[1], Sam P. Jones[1], Laura Clavé[1], Lisa Wingate[1], Teresa E. Gimeno[1,2,3], Bastien Fréjaville[1], Steve Wohl[1], Jérôme Ogée[1]

[1]INRA, UMR ISPA, F-33140, Villenave d'Ornon, France
[2]BC3 - Basque Centre for Climate Change - Klima Aldaketa Ikergai, E-48940, Leioa, Spain
[3]IKERBASQUE, Basque Foundation for Science, 48008 Bilbao, Spain

*Correspondence to*: Adrià Barbeta (adria.barbeta-margarit@inra.fr) and Jérôme Ogée (jerome.ogee@inra.fr)

**Abstract.** We investigated plant water sources of an emblematic refugial population of *Fagus sylvatica* (L.) in the Ciron river gorges in South-Western France using stable water isotopes. It is generally assumed that no isotopic fractionation occurs during root water uptake, so that the isotopic composition of xylem water effectively reflects that of source water. However, this assumption has been challenged by recent studies that found that plant water did not reflect any mixture of the potential water sources at some dates during the growing season. In this context, highly resolved datasets covering a range of environmental conditions could shed light on possible plant-soil fractionation processes responsible for this phenomenon. In this study, the hydrogen ($\delta^2$H) and oxygen ($\delta^{18}$O) isotope compositions of all potential tree water sources and xylem water were measured fortnightly over an entire growing season. Using a Bayesian isotope mixing model (MixSIAR), we quantified the relative contribution of water sources for *F. sylvatica* and *Quercus robur* (L.) trees. The $\delta^{18}$O data alone indicated that both species used a mix of top and deep soil water over the season, with *Q. robur* using deeper soil water than *F. sylvatica*. The contribution of stream water appeared to be marginal despite the proximity of the trees to the stream, as already reported for other riparian forests. Xylem water $\delta^{18}$O could always be interpreted as a mixture of deep and shallow soil waters, but the $\delta^2$H of xylem water was often more depleted than the considered water sources. We argue that an isotopic fractionation in the unsaturated zone and/or within the plant tissues could underlie this unexpected relatively depleted $\delta^2$H of xylem water, as already observed in halophytic and xerophytic species. By means of a sensitivity analysis, we found that the estimation of plant water sources using mixing models was strongly affected by this $\delta^2$H depletion. A better understanding of what causes this isotopic separation between xylem and source water is urgently needed.

# 1 Introduction

## 1.1 Why is an improved understanding of tree water use needed?

On-going climate change, through the combination of altered precipitation regimes and warmer temperatures, is affecting terrestrial ecosystems globally, promoting rapid and widespread changes in forest cover (e.g. Allen et al., 2015). This is because soil-plant interactions and species-specific water use can provoke shifts in forest species' distributions through readjustments in species abundance (Clark et al., 2016). In this context, it is a priority to better understand how spatial and temporal dynamics of water use by trees will be affected in the future. This will not only reduce uncertainties in the projections of forested areas (e.g. Cáceres et al., 2015; Good et al., 2017), but will also improve our ecohydrological understanding of biosphere-atmosphere feedbacks and associated climate change (Berg et al., 2016; Seneviratne et al., 2013). This may also help understand how climate refugia facilitate the persistence of important biodiversity hotspots (McLaughlin et al., 2017).

## 1.2 The isotopic tracing method to study tree water use

Water stable isotopes ($\delta^{18}$O and $\delta^2$H) are commonly used as tracers of plant water use (Dawson et al., 2002; Rothfuss and Javaux, 2017, see also references below). This requires sampling all potential water sources as well as xylem water and analysing these waters for isotopic composition. Because plants can access various pools of water belowground (soil water at different depths, ground water, rock water) but also at the soil surface (recent rain, river water), on leaf surfaces (dew) or even in the air (fog, water vapour), sampling all potential water sources can be technically challenging, destructive, expensive and/or time-consuming, and this may hamper the assessment of their temporal and spatial variability within river catchments (Fahey et al., 2017). The development of laser-based isotopic analysers in the last decade has however increased the throughput of water isotope measurements, providing opportunities to carry out observational studies at higher temporal and spatial resolution (Martín-Gómez et al., 2015).

The isotope tracing methodology is based on two main principles. Firstly (H1), it is assumed that isotopic fractionation during root water uptake (and/or xylem water redistribution) does not occur (Allison et al., 1983; Dawson and Ehleringer, 1991; Ehleringer and Dawson, 1992; White et al., 1985). Some recent studies have challenged this assumption by showing evidence of isotopic fractionation during root water uptake, but it was suggested that such an isotopic fractionation was a specific feature of saline or xeric environments (Ellsworth and Williams, 2007; Lin and Sternberg, 1993). Secondly (H2), it is essential that the isotopic compositions of all potential water sources are different enough to distinguish their relative contribution to xylem water (Ehleringer & Dawson, 1992). Processes underlying the variability in source water isotopic composition include the temporal variability in the isotopic composition of rainfall and mixing processes of water in the subsurface (Allison and Hughes, 1983; Brooks et al., 2010; Tang and Feng, 2001), the evaporative enrichment of water in surface soil layers (Allison, 1982; Sprenger et al., 2016; Tang and Feng, 2001), the seasonality of groundwater and rock moisture recharge (Oshun et al., 2015) or isotopic processes during fog water droplet formation (Scholl et al., 2011). There is

no certainty however that these processes will necessarily lead to different isotopic compositions of all potential water sources. Still, if H1 is true, the $\delta^{18}O$ and $\delta^2H$ of xylem water should always lie within the range of values of all water sources.

**1.3 Possible caveats of the isotopic tracing method**

The water isotope tracing technique has succeeded in advancing our understanding of plant water uptake (Ehleringer and Dawson 1992; Dawson et al. 2002). However, it occasionally leads to results that are rather unexpected. For instance, a pioneering study using hydrogen isotopes alone concluded that mature streamside riparian trees in a semi-arid catchment did not use stream water but were dependent on an unidentified, relatively more depleted, water source, hypothesized to reflect groundwater (Dawson and Ehleringer, 1991). On the other hand, smaller trees seemed to rely on stream water. Another study

conducted in a seasonally dry conifer forest in Oregon found that the $\delta^{18}O$ and $\delta^2H$ of soil and tree water were similar during the dry season, and clearly distinct from stream water, even when sampled near the stream (Brooks et al. 2010). This led to the two water worlds (TWW) hypothesis whereby the first depleted rainfall water after a rainless summer fills small soil pores first and does not contribute to river flow nor to mixing with subsequent rain events, as it was already observed in ecosystems with less seasonality of rainfall (Tang and Feng, 2001). This water pool eventually participates in soil evaporation but mainly

remains in the soil until being used by plant transpiration during the following dry summer. In light of this TWW hypothesis, Bowling *et al.* (2017) revisited the study site of Dawson and Ehleringer (1991) many years later and re-measured hydrogen but also oxygen water isotopes in xylem water, soil water at different depths and stream water, in addition to (this time) groundwater and snowmelt water. They suggested that, if the TWW were true, the soil, still dry at the end of winter, should get recharged in spring during snow melt, leading to depleted snowpack water being locked in small soil pores and used by

the trees later in the summer. Although the vertical distribution of soil water isotopes following snowmelt seemed consistent with the TWW hypothesis, neither snow melt water, nor groundwater could be identified as an alternative source for riverside trees. They concluded that the dual isotope approach could not unambiguously determine the water sources of these riparian trees but that soil moisture seemed to be the most likely candidate, despite the proximity of the river. Oerter et al. (2019) revisited once more the same site but analysed the isotopic composition of soil water at various depths using the same technique

as in previous studies (i.e. collecting small soil samples and extracting their water under vacuum in the lab) but also using vapour probes and assuming liquid-vapour isotopic equilibration. The isotopic composition of soil water using both techniques was undistinguishable but also, this time, was found similar to that of xylem water, which led the authors to conclude that there was not a missing source. However, they could not find an explanation for the observed isotopic separation found in previous years/studies but noted that the isotopic composition of rainwater (and thus probably the precipitation amounts of each rain

event) was quite different between 2014 (the year studied in Bowling *et al.* 2017) and 2016 (the year studied in Oerter et al. (2019). They concluded that "while [they] were able to identify the so-called "missing water source" for these trees using this combination of measurement methods, the mechanisms that create co-existing soil water pools with distinct isotopic compositions, and thus which may lead to ecohydrologic separation are still not clear".

**1.4 Rock moisture as an alternative plant water source?**

Plant water source studies in which the xylem water isotopic composition does not spread within the range of the sources' isotopic compositions often acknowledge that a relevant source of water may not have been sampled (Bowling et al., 2017; Geris et al., 2017). Not many studies sample rock moisture, but it has been shown that plants growing on rocky outcrops rely on water stored in the bedrock. The contribution of bedrock water to plant transpiration in such sites is comparable to that of groundwater in sites with more developed soils and a shallow water table (Barbeta & Peñuelas, 2017 and refs therein). Indeed,

rock moisture ('the exchangeable water stored in the matrix and fractures of weathered bedrock') can represent up to 27% of annual rainfall, and can be taken up by trees during the dry season (Rempe and Dietrich, 2018). Moreover, the water stored in soil rock fragments can have an isotopic composition distinct to that of soil water or groundwater, being either relatively more depleted (in the case of $\delta^2H$ in Oshun et al., 2015), or more enriched (Palacio et al., 2014; Rong et al., 2011). Such variable and contrasted isotopic effects of lithology are to be expected for differing minerals, and can even cause fractionations of

opposite signs for the hydrogen and oxygen isotopes (Meißner et al., 2014; Oerter et al., 2014). Thus, wherever weathered rocks constitute a large fraction of the soil volume, the isotopic composition of rock moisture should be measured as rock moisture could constitute a significant alternative plant water source.

**1.5 Evidence of isotope fractionation during root water uptake**

Although it cannot be ruled out that rock water in the carbonate-rich soil of Bowling et al. (2017) was a significant source of

water for trees or caused any unexpected isotope effects, the very clayey soil texture reported by Brooks et al. (2010) seems less likely to contain a large rock water component. Oerter et al. (2014) showed that cations adsorbed to clay minerals create isotopically organized hydration spheres of water around them and thereby sequester these water molecules away from the bulk water. However, even if the majority of the water contained in small pores is adsorbed water that does not interact with the more mobile water (the TWW hypothesis), in summer, when only water in small pores is accessible to the trees, there

should be an isotopic match between soil pore and xylem water, unless (H1) is not true and isotopic fractionation occurs during root uptake. In this context, a recent controlled experiment conducted on potted avocado (*Persea americana*) trees has revealed isotopic fractionation during root water uptake in non-saline, relatively moist environments (Vargas et al. 2017), clearly questioning the validity of (H1). Vargas et al. (2017) showed that *P. americana* plants discriminated against hydrogen isotopes about 10 times more than oxygen isotopes during water uptake, and this discrimination increased with soil water loss, porosity

and particle size. Interestingly, the dataset reported by Brooks et al. (2010) contain a substantial number of xylem water samples that occupy the $\delta^{18}O$-$\delta^2H$ space well below the soil water line, suggestive of deuterium fractionation processes during root water uptake. In fact, a growing number of studies are reporting xylem water with an isotopic composition that is relatively depleted to that of the considered source**s** (De Deurwaerder et al., 2018; Evaristo et al., 2017; Geris et al., 2017; Oerter and Bowen, 2019; Wang et al., 2017), suggesting that isotopic fractionation during root water uptake may be more common than

previously thought. If such fractionation processes are not taken into account, the estimation of plant water sources may be

miscalculated. The effect of an eventual deuterium separation between soil and xylem waters on the quantification of plant water sources was recently addressed by Evaristo et al. (2017). They showed that erroneous results could be obtained when a simple mass balance approach using hydrogen isotopes only was implemented, but also concluded that results were less sensitive to deuterium fractionation when both deuterium and oxygen isotopes were combined within a Bayesian inference approach (Evaristo et al. 2017).

### 1.6 Aim of the study

The aim of this study was to identify the water sources of a refugial population of *Fagus sylvatica* (L.) in SW France, nearing the southernmost distribution limit of this species. Evidence from studies of population genetics (De Lafontaine et al., 2013) and *in situ* soil macrofossil charcoals radiocarbon-dated back to more than 40 kyr before present, when the area was a periglacial desert (de Lafontaine *et al.*, 2014), indicate that the Ciron valley acted as a climate refugia during the Last Glacial Maximum (de Lafontaine et al., 2014; Timbal and Ducousso, 2010). Thereafter *F. sylvatica* expanded northwards and colonized the areas of its current distribution range from this and other populations in Southern Europe (Gavin et al., 2014). The population is hypothesised to have persisted there since the Last Glacial Maximum (de Lafontaine *et al.*, 2014) because of an array of edaphic, thermal and hydric features decoupled from the surrounding regional environment, notably: convergent topography, frequent fog, short distances to a stream and complex lithology. In an attempt to understand better the ecohydrological mechanisms shaping this refugium, we sampled potential source waters as well as xylem water, analysed their isotopic composition and applied isotope mixing models to quantify the relative contribution of different water sources to both *F. sylvatica* and the more regionally widespread *Quercus robur* (L.). To do so, we also addressed a number of the caveats raised above including the sampling of all potential water sources, including fog and rock water, in addition to measuring both water isotopes to identify better possible isotopic fractionation during root water uptake. We used a Bayesian inference approach to quantify how plant source water varied seasonally, between species, and with distance to the river. In parallel to the ecological focus of our study, the reported isotopic dataset spanning a whole growing season was also used to explore the potential effect of isotopic fractionation on the quantification of tree water sources.

## 2. Methods

### 2.1 Study site and experimental design

European beech (*Fagus sylvatica* L.) is a deciduous broadleaved tree species distributed across most of Western and Central Europe. The population that is the focus of this study is found along a mixed riparian forest on the karstic canyon formed by the Ciron, a tributary of the Garonne river, in Gironde, a south-western French region (44°23 N, 0°18 W, 60 m a.s.l.). The soil there has a fine texture and is slightly less organic than the sandy soils found in the surroundings, typical of the Aquitaine basin (Table S1). Importantly, the presence of limestone rocks weathered to various degrees creates a distinguishable carbonate-rich C horizon between 50 and 120 cm belowground (Table S1). Interestingly this European beech population is

restricted either to the sheltered Ciron ravine or to slightly more distant sites (100m) located on irregular microtopography with small karstic depressions. In this riparian forest *F. sylvatica* trees coexist with other deciduous species such as *Quercus robur* L., a regionally common tree species that dominates the canopy further away from the river. Other tree species within the riparian forest are *Carpinus betulus* L., *Alnus glutinosa* L., *Corylus avellana* L. and *Tilia platyphyllos* Scop. At the riparian forest limits beyond the chalky soil areas, we find plantations of *Pinus pinaster* Ait.,clear cuts or agricultural fields.

The studied area has a temperate oceanic climate (Cfb in the Köppen-Geiger classification). Daily meteorological data was available from a weather station located at about 20 km from the studied site, and long-term (1897-present) monthly temperature and precipitation data was also available from another weather station located 16 km away from the studied area. Streamflow data was obtained from a stream gauge located about 4 km downstream of our study area. Over the period 1897-2015, the mean annual temperature was 12.9°C and the mean annual precipitation was 813 mm y$^{-1}$, distributed rather evenly over the season. Since 1897, the mean annual temperature has increased by +1.0°C ($P < 0.001$), whereas precipitation has not showed any trend.

Early in 2017, three field plots with different conditions were set up within the riparian forest. Two of the plots were located on opposite sides of the river (NE and SW) to explore exposition effects, and a third plot, adjacent to an open area formerly occupied by a *P. pinaster* plantation, was chosen to explore the effect of forest fragmentation on the microclimate and notably the fog occurrence. In each of the plots, we selected five mature *F. sylvatica* and three *Q. robur* individuals of 80-150 years and all occupying dominant positions in the canopy. In addition, we selected six non-dominant *F. sylvatica* trees in two of the plots to explore the effect of tree size (Dawson & Ehleringer 1991, 1993). The maximum distance between trees from the same plot was 15 m. All selected trees were sampled fortnightly from mid-April to early November 2017. In order to measure the xylem water isotopic composition, several twigs were collected from every tree, rapidly peeled to remove bark and phloem, then placed in an air-tight Exetainer® sealed with Parafilm® and kept in a cool box until they were stored in the lab at 4°C. For four trees (three *F. sylvatica* and one *Q. robur*), the canopy could not be accessed and so we extracted xylem samples from coarse roots with an increment borer. Three soil cores per plot, located randomly amongst the sampled trees, were extracted with a soil auger. Each soil core was split into top soil (0-10 cm) and deep soil (from 70-80 to 110-120 cm depending on the depth of the bedrock). The isotopic composition of soil water varies strongly with depth and time (Allison et al. 1983) The two soil sampled layers here were considered the best representatives of evaporation-exposed shallow soil layers and deeper ones only affected by infiltrating water and subsequent mixing. This decision was based on the characteristics of the soil profile (Table S1), in which the sampled shallow and deep soil layer had a texture that could hold most of the soil moisture, compared to middle soil layer (20-40cm) that has a very coarse texture and lower water holding capacity. Soil samples were placed in 20 mL vials with positive insert screw-top caps, sealed with Parafilm® and kept in a cool box until they were stored in the lab at 4°C. From July onwards, we also sampled limestone rocks. We dug horizontally into rocky edges to avoid the effect of evaporation, and collected one sample per plot and sampling date.

In addition to soil, xylem and rocks, we collected for every sampling date water from the stream, groundwater from a well located *ca*. 50 m from the river, and fog and rain water from collectors installed in a small open area about 100 m away

from one of our plots. Both rain and fog collectors were connected *via* a funnel to a thermally insulated water reservoir buried in the ground with minimal contact with the open air following the recommendations of the GNIP network ([http://www-naweb.iaea.org/napc/ih/IHS_resources_gnip.html](http://www-naweb.iaea.org/napc/ih/IHS_resources_gnip.html)). Each rain and fog water sample corresponds to the averaged (amount-weighted) value of the water that precipitated since the previous sampling date. The local meteoric water line (LMWL) was constructed with rain water isotope data collected monthly since February 2007 at a Global Network of Isotopes in Precipitation (GNIP) station located in Cestas, France (Fig. 2). The fog collector was custom-built following the design of the single-stage Caltech Active Strand Cloud water Collector (CASCC2, Demoz et al. 1996). This design has been shown to be well suited for water isotope studies (Spiegel et al. 2012). According to the theory presented in Demoz et al. (1996) our one-stage fog collector is ill-designed to collect small fog events (i.e. clouds with droplet sizes of 7 µm or less, corresponding to a liquid water content of less than 0.01 g m$^{-3}$) but such fog events are unlikely to have any significant contribution to the water source of the trees (i.e. less than 0.5 L h$^{-1}$ tree$^{-1}$, assuming a surface of exchange of 60 m$^2$ tree$^{-1}$ and an average wind speed through the tree crown of 0.2 m s$^{-1}$ during such fog events). On the other hand, any fog event with enough water made up of droplets larger than 7µm in diameter was (in theory) collected, and the isotopic composition of this fraction of the cloud was expected to be representative of the entire cloud, because the isotopic composition of cloud droplets is independent of their size (Spiegel et al., 2012).

## 2.2 Water extraction and determination of stable isotope composition

The water contained in soil, xylem and rock samples was extracted using a cryogenic vacuum distillation system based on the design and methodology described by Orlowski *et al.* (2013). A detailed description of the system used is available in Jones *et al.* (2017). Briefly, the pressure in the extraction line was set at less than 1 Pa at the start of the extraction (i.e. when the samples were still frozen in liquid nitrogen). The samples were then gradually (within 1h) heated up to 80°C (soils) or 60°C (xylem) for 2.5 hours. The pressure line was continuously recorded using sub-atmospheric pressure sensors (APG100 Active Pirani Vacuum Gauges, Edwards, Burgess Hill, UK) to check that the lines remained leak-tight throughout the entire extraction. Gravimetric water content was assessed for each sample using the sample weight before and after water extraction. We also checked that the water extraction had been completed by oven drying all samples at 105°C for 24h and re-weighing them.

The isotopic composition ($\delta^2$H and $\delta^{18}$O) of the different waters were measured with an off-axis integrated cavity optical spectrometer (TIWA-45EP, Los Gatos Research, USA) coupled to an auto-sampler. Details on the processing and post-correction of water samples can be found in Jones *et al.* (2017). The presence of organic compounds (ethanol, methanol and/or other biogenic volatile compounds) in water samples can lead to large isotopic discrepancies in laser-based analyses (Martín-Gómez *et al.*, 2015; Wassenaar et al. 2018). Organic compounds are found in certain soil types (Orlowski et al., 2018), but are more typically found in water extracted from plant tissues (Zhao et al. 2011). Post-corrections to account for the presence of organic compounds in water can be applied, based on metrics from the measured absorption spectrum (Brian Leen et al., 2012; Schultz et al., 2011). Nonetheless, these post-processing functions must be developed for each individual instrument. Following Schultz *et al.* (2011), we thus developed our own post-corrections by analysing milliQ waters mixed with methanol

and/or ethanol at various concentrations and by fitting the measured deviation from the expected isotope ratio to the narrow- and broad-band metrics provided by the instrument. We verified the performance of our correction with the contaminated standard WICO5 (Wassenaar et al. 2018). Xylem water samples generally exhibited higher narrow and broad band metrics compared to rain or even soil water samples but the corrections on xylem samples were always quite small (ca. 1.5 ‰ for $\delta^2H$ and 0.7 ‰ for $\delta^{18}O$) compared to the correction we had to apply on the WICO5 sample, or on some of our water-alcohol mixtures used to derive our in-house post-processing functions. All isotopic data reported here are expressed on the VSMOW-SLAP scale.

## 2.3 Data analysis

The relationships between xylem water and its potential sources were compared at the plot level. All the analyses described below are also calculated at the plot level. Because no significant difference was found between the isotopic compositions of xylem (or water sources) between the different studied plots, "plot" was set as a random factor. To assess whether there was an isotopic offset between tree xylem water and its potential sources, the concept of the line-conditioned excess (LC-excess) proposed by Landwehr & Coplen (2006) was used: LC-excess = $\delta^2H – a\,\delta^{18}O - b$, where $a$ and $b$ correspond to the slope and intercept of the Local Meteoric Water Line (LMWL), respectively. However, because the source water for a tree is more likely to be made of soil water than rain water directly, we modified the equation above and computed the deviation of a given xylem water with respect to the soil water line (SW-excess) from the same plot and date:

$$SW\text{-excess} = \delta^2H – a_s\,\delta^{18}O - b_s,$$

where $a_s$ and $b_s$ are the slope and intercept of the soil water line for a given plot and date, respectively, and $\delta^2H$ and $\delta^{18}O$ correspond to the isotopic composition of a xylem water sample collected on that plot at that date. The slope and intercept $a_s$ and $b_s$ were computed by performing a linear regression on all the soil water isotope data from the surface and deep horizons collected at a given plot and date. The SW-excess of xylem water is an indicator of the $\delta^2H$ offsets between xylem samples and their corresponding soil water lines. Positive SW-excess values indicate xylem samples that are more enriched in deuterium than the soil water line (and are thus positioned above soil water in a $\delta^{18}O$-$\delta^2H$ diagram), while negative SW-excess values indicate xylem samples that are more depleted in deuterium than the soil water line (and are thus positioned below soil water in a $\delta^{18}O$-$\delta^2H$ diagram).

The contribution of different water sources to that of xylem water was estimated using the *MixSIAR* package (Stock and Semmens, 2016) in *R* (R Core Development Team, 2012). Different mixing models were ran in the script version of the package, and the number of Markov chain Monte-Carlo iterations was increased manually (by trial and error) until convergence was reached and the results for the Gelman and Geweke diagnostics were acceptable. We grouped together trees of the same plot, species and date altogether, and thus specified the residual error term in the isotope mixing models (Parnell et al., 2010). The potential tree water sources that we considered were restricted to the top and deep soil water and stream/groundwater. Stream and ground waters were pooled together as they were isotopically indistinguishable. Fog and rock moisture were not

included as potential water sources with this approach because their isotopic signatures were very distant to xylem waters in a $\delta^{18}O$-$\delta^{2}H$ diagram, and because there were only a limited number of campaigns when they were measured. In order to test the sensitivity of *MixSIAR* to different data inputs, the models were run with four different input data; (1) $\delta^{2}H$ and $\delta^{18}O$, (2) $\delta^{2}H$ and $\delta^{18}O$ after subtracting the SW-excess from the $\delta^{2}H$ of xylem samples, (3) only $\delta^{18}O$ and (4) only $\delta^{2}H$. Correcting xylem 260  $\delta^{2}H$ with SW-excess implies that tree water uptake relies only on soil water pools because the SW-excess is calculated using the slope and intercept of the soil water line. However, the lower part of this line usually overlaps with unenriched stream/ground water. Thus, we expected that $\delta^{2}H$ departures from this line are meaningful in potential cases where trees are accessing not only soil water but also stream water.

The spatial, temporal, species-specific and size-related statistical comparisons between the isotopic compositions of 265  grouped samples were analysed using linear models or, when plot and date were necessarily set as random factors, linear mixed models from the package *lme4* (Bates et al., 2015) in R. For instance, for comparisons between groups across several dates, the date of sampling was set as a random factor. In order to understand the factors driving the observed SW-excess of xylem water, we fitted Generalized Linear Mixed Models (GLMM) including soil moisture (top and deep), type of sampling (coarse root or branch) soil water isotopes, tree diameter (DBH), as well as rainfall and vapour pressure deficit (VPD) prior to sampling 270  and using the tree species as an explanatory variable. We selected the best model by means of the second-order Akaike Information Criteria (AIC). Given the water source contributions estimated with *MixSIAR* using different input data, we assessed their correlations with top and deep soil moisture, rainfall and VPD, also using GLMM from *lme4*.

## 3. Results

### 3.1 Environmental conditions

275  The mean temperature of the 2017 (April-November) growing season was 0.4°C warmer than the long-term average, but 0.5°C cooler than the average of the last 25 years. Precipitation during the 2017 growing season was 20% lower than the long-term average but close to the average of the last 25 years. There was a clear deficit in precipitation from the previous winter (estimated from December 2016 to March 2017) that caused a 43% reduction in streamflow compared to the 2000-2017 average, throughout the entire growing season (Fig. 1). Deep soil layers progressively dried over the entire growing season up 280  to the last sampling campaign in November, while top soil moisture was usually higher but also more variable, with relatively high levels at the beginning and end of the season and levels as low as the deep soil layer only in mid-summer (Fig. 1). Based on the water retention properties of top and deep soil layers, we estimated that the permanent wilting point was reached in the top soil only in early September, and from late July to the end of the season in the deep soil. Using a rock density of 2.5 g cm$^{-3}$, we estimated the mean volumetric water content of limestone rocks to be around 12%, which is comparable to that of the deep 285  soil.

### 3.2. Stable isotopic composition of tree water sources

The long-term (2007-present) local meteoric water line (LMWL) using the closest GNIP station (see Material and Methods) is shown in each panel of Fig. 2. The rain data collected fortnightly plotted closely to this line and ranged, from -7.0‰ to -1.3‰ in $\delta^{18}O$ and -46.8‰ to -5.4‰ in $\delta^2H$ (Fig. 2). Fog water ranged from between -6.5‰ and -0.9‰ in $\delta^{18}O$ and between -32.4‰ and -8.4‰ in $\delta^2H$ (Fig. 2) and was not significantly different from rain water (P>0.05 for both isotopes, Fig. S1). Stream and groundwater had isotopic compositions that were not statistically different and very stable over time (-5.9 ± 0.2‰ in $\delta^{18}O$ and -36.8 ± 0.8‰ in $\delta^2H$).

Soil water samples occupied the $\delta^2H$-$\delta^{18}O$ space on the right side of the LMWL (Fig. 2). On average over the growing season, top soil water was significantly more enriched than deep soil ($P < 0.001$ for both isotopes) as a result of evaporative enrichment at the soil surface. The resultant soil water line (SWL) had a mean slope of 5.17 (ranging from 4.01 to 9.99 depending on the sampling date), which is significantly smaller than the slope of the LMWL (6.73). The difference in $\delta^{18}O$ between top and deep soil water was significantly smaller in the plot within a mixed broadleaved forest ($P < 0.05$), suggesting that soil evaporation was probably lower at this plot.

Over the season, rainfall amounts over the 15 days preceding each sampling campaign had a negative effect on top soil water $\delta^{18}O$ and $\delta^2H$ ($P < 0.001$) and no significant effect on the isotopic composition of the deep soil water, typical of shallow infiltration-evaporation cycles (Barnes and Allison 1988). In the top soil, water content was negatively correlated with soil water $\delta^{18}O$ (P < 0.05), but not with $\delta^2H$. This is surprising because isotopic fractionation occurring during soil water evaporation and water vapour and liquid diffusion should affect both water isotope signals in the same direction. The fact that these water signals respond differently to top soil water content but similarly to rainfall amount (see above) indicates that observed changes in top soil water isotope signals are primarily governed by the isotopic composition of the precipitation input and only secondarily by soil water evaporative enrichment. It may also be that hydrogen isotope of soil water are reflecting extra fractionation processes (e.g. root uptake) compared to their oxygen isotope counterparts. No similar correlation was observed between soil water content and $\delta^{18}O$ (or $\delta^2H$) in the deep soil probably because the range of variations was smaller (Fig. 1 and Fig. 2). Finally, $\delta^2H$ and $\delta^{18}O$ of rock moisture were significantly more enriched than those of top and deep soil water, but fell along the LMWL (Fig. 2). The isotopic signal of rock moisture did not differ between plots over time, nor did it correlate with weather conditions or with the isotopic signal of top or deep soil water, and thus rock soil water isotopic composition was excluded from further analyses.

### 3.3 Stable isotopic composition of xylem water

The isotopic composition of xylem water always fell underneath the LMWL in the dual-isotope space (Fig. 2). Xylem water from the first campaign on the 19th of April (i.e. just before or during budburst), was exceptionally enriched (Fig. 3) and fell in the upper right part of the dual-isotope space (top left panel in Fig. 2), except for those trees that had already flushed their

leaves. This could be indicative of stem evaporative enrichment over winter, as observed in other species (Bowling et al., 2017; Martin-Gomez et al., 2017). Excluding this first campaign, xylem water samples of both *F. sylvatica* and *Q. robur* overall had a more depleted $\delta^2H$ than top and deep soil water ($P < 0.001$), as illustrated by Fig. 3. Consequently, a large number of the xylem samples fell outside the range of the considered sources in the dual-isotope space (Fig. 2).

The diameters at breast height of trees (DBH) were negatively correlated with both isotopes of xylem water samples ($P < 0.001$). Consequently, dominant trees of *F. sylvatica* had more depleted xylem water than non-dominant trees ($P < 0.01$ for both isotopes). Xylem water from *F. sylvatica* trees presented marginally more enriched values of $\delta^{18}O$ ($P < 0.05$) and $\delta^2H$ ($P < 0.1$) than *Q. robur* trees. To our surprise, no significant differences were found in xylem water isotopes between the three studied plots. The four trees (all on the same plot) in which xylem water was extracted from outcropping coarse roots (rather than from twigs) showed a significantly more depleted $\delta^2H$ over the whole season ($P < 0.001$), but no significant difference in $\delta^{18}O$, compared to all the other trees (Fig. 4). The $\delta^2H$ offset between xylem and soil water samples still persisted after excluding these coarse root samples, demonstrating that xylem water $\delta^2H$ exhibited different patterns than $\delta^{18}O$.

The isotopic offset between xylem and soil water samples was assessed by calculating the SW-excess. On average, xylem water samples had a SW-excess of -8.40 ± 5.37‰. There were no significant differences in xylem SW-excess between species, and its seasonal variations were small (Fig. 5). Although canopy position and DBH had no effect on the SW-excess, the type of sampling had a strong influence, as the SW-excess was significantly more negative in trees whose xylem water had been sampled from coarse roots as opposed to twigs (Fig. 4, Table S2). The linear regression of the soil water line was significant for most sampling dates and plots (Table S3). Non-significant regressions may be caused non-monotonic isotopic soil profiles, in which intermediate layers could be relatively more depleted in both isotopes than surface soil layers. In such cases, the estimation of the SW-excess could be less meaningful regarding isotopic offsets between xylem and soil water. Consequently, we removed from the multivariate analysis of the SW-excess data corresponding to sampling dates and plots with non-significant soil water line regressions. Still, the SW-excess did not significantly differ between cases with significant soil water lines and cases with non-significant soil water lines ($P = 0.45$).

The GLMM used to understand the factors driving the SW-excess across time and space explained 33.3% of the variance (Table S2). The model that best fitted the data showed that top soil water content had a positive effect on SW-excess, whereas rainfall, top soil water $\delta^2H$ and daytime VPD had negative effects (larger isotopic offset between xylem water and the soil water line). The variables with larger relative importance were the type of sampling (coarse roots or twigs, see Fig. 4) and rainfall of the week prior to the sampling date.

### 3.4. Isotopic mixing models

The potential tree water sources that we considered were restricted to the top and deep soil water and stream/groundwater. Stream and ground waters were pooled together as they were isotopically indistinguishable (Fig. 2). Fog and rock moisture

were not included as potential water sources because their isotopic signatures were very enriched compared to xylem water but also soil and stream/ground water (Fig. 2), so that their contribution would have moved xylem water samples above and to the right of the other potential sources in the dual isotope plot, i.e. the opposite of what we observed. Also fog water could only be collected at the end of the summer, so is unlikely to have been a significant source of water in either spring or early summer. The first set of isotopic mixing models were only run for the dominant trees of *F. sylvatica* and Q. *robur* using both $\delta^{18}O$ and $\delta^2H$ data. Because non-dominant trees were only sampled for *F. sylvatica*, and not for *Q. robur*, we preferred to exclude them when comparing the two species. On average, these mixing models indicated that *F. sylvatica* trees used a mix of top and deep soil water, with a marginal contribution of stream water (Fig. 6). The same mixing models also indicated that *Q. robur* relied mostly on soil water as well, but had significantly higher contributions from stream (P < 0.01) and deep soil water (P < 0.01), and consequently lower contributions from top soil water (P < 0.001), compared to *F. sylvatica*. Nonetheless, both species followed similar temporal patterns (Fig. 6). The non-dominant *F. sylvatica* trees also had similar source contributions as the dominant ones, although with a slight but surprisingly higher relative uptake from stream water (Fig. S2). Differences between plots were not significant (not shown).

In a second step, we focused on the sensitivity of the isotopic mixing models to the observed $\delta^2H$ offset and the dual- versus single-isotope approach. For this, we only used the isotopic data for dominant *F. sylvatica* trees (N = 15). Correcting values of xylem $\delta^2H$ for their SW-excess significantly affected the estimated source contributions of *F. sylvatica* (Fig. 7). The dual isotope model with corrected $\delta^2H$ values estimated a higher contribution of stream water late in the season ($P < 0.001$) and deep soil water in the summer compared to the dual isotope model with the original $\delta^2H$ values. This was naturally accompanied by a reduction in the contribution of top soil water during summer. The single-isotope approach using only $\delta^{18}O$ also estimated a higher contribution of stream water ($P < 0.001$), following closely that of the deep soil, and a lower contribution of top soil water compared to the dual-isotope approach with uncorrected $\delta^2H$ (Fig. 7). On the other hand, a single-isotope approach using only $\delta^2H$ led to very similar contribution patterns as the dual-isotope approach with uncorrected $\delta^2H$, except at the very beginning and end of the growing season (Fig. 7).

The discrepancy in the estimation of source contribution to xylem water of isotope mixing models with different input data also translated into a contrasting relationship with environmental data (rainfall, VPD and soil moisture). These relationships are reported in Table 1, separated by source and input data. Overall, the models using a dual-isotope approach but with corrected $\delta^2H$ values, or only $\delta^{18}O$ showed the strongest and most plausible correlations with environmental variables over the growing season. Although the contribution of stream water to xylem water estimated from $\delta^2H$ only led to the best correlations with rainfall amounts and VPD, the sign of these correlations was the opposite of what is expected. That said, the use of only one isotope was not sufficient to disentangle the contribution of various water sources for some campaigns where the isotopic compositions of the different water sources were too similar (Fig. 8). In these cases, the Bayesian mixing models predicted equal contributions for each of the three water sources considered (e.g. on the 4[th] of July for $\delta^{18}O$ only, Fig. 7).

## 4. Discussion

### 4.1 Potential causes for the $\delta^2H$ offset between xylem water and source water

Our results support those from recent studies reporting xylem water with a hydrogen isotope ratio more depleted than any of the considered water sources, and thus of any of their combinations (Evaristo et al., 2017; Geris et al., 2017; Oerter and Bowen, 2019). The diversity of methodologies used for the extraction of waters and their isotopic determination in all these studies, including ours, minimises the likelihood of a common analytical or methodological bias. Furthermore, isotopic offsets measured between xylem and source water were consistent over time and space (Fig. 2 and Fig. 5). Other field datasets have shown similar isotopic offsets in semi-arid (Dawson and Ehleringer, 1991; Oerter and Bowen, 2019; Zhao et al., 2016) and saline (Lin & Stenrberg 1993) environments, but here we show that it also occurs and persists in temperate deciduous trees growing in a mild oceanic climate. Furthermore, isotopic offsets between xylem water and soil water in potted plants (Ellsworth & Stenberg 2007; Vargas *et al.*, 2017) and plants in botanical gardens (Evaristo et al., 2017) have also been reported and discussed to some extent. In addition, studies from tropical (De Deurwaerder et al., 2018), semi-arid (Wang et al., 2017), temperate (Bertrand et al., 2014; Brooks et al., 2010) and northern ecosystems (Geris et al., 2015, 2017) have also reported offsets similar in magnitude to those observed in our study. However, these results and their repercussions for partitioning were not fully discussed nor explored. Our results show that $\delta^2H$ offsets between xylem water and source water complicate the identification of plant water sources and the source contributions estimated by Bayesian isotopic mixing models (Fig. 7), a finding in contrast with recent studies (Evaristo et al., 2017).

The mismatch between xylem water and source water isotopes may be caused by three non-exclusive processes: (1) a water isotope separation between bound and mobile soil water (Tang and Feng, 2001; Brooks et al. 2010), (2) a water isotope fractionation occurring at the soil-root interface (Ellsworth & Stenberg 2007; Vargas et al. 2017) or (3) a water isotope compartmentalisation between vessel water and other stem water pools (Zhao et al. 2016). In particular, surface-water interactions operating at the pore level (Oerter & Bowen, 2017) and varying as a function of particle size (Gaj et al., 2017) or cation content (Oerter et al., 2014) may create isotopic heterogeneity within the soil matrix. The soil in our study site is sandy, thus the effect of interactions with clay-absorbed cations are likely to be small (Fig. S2). However, surface-water interactions on quartz silica or carbon-rich materials have also been shown to affect the water isotope composition of adsorbed water (Richard et al. 2007; Lin & Horita 2016, 2018; Chen et al. 2016). The same process seems to also happen in real soils. Indeed, Oerter and Bowen (2017) compared the soil water isotopic composition of mobile water (by liquid-vapour equilibration) and total water (by destructive soil sampling and vacuum extraction, i.e. including both mobile and adsorbed water) and found some disagreement between the two types of water depending on soil texture and water content and concluded that mobile and total soil waters may be isotopically separated. A disparity between total soil water and more mobile water accessible to the plant would create a mismatch between plant and bulk soil water. However, adsorbed water is expected to be more depleted than bulk soil water (Oerter and Bowen 2017; Lin et al., 2018; Lin and Horita, 2016), so the more mobile water taken up by the plants should be more enriched than bulk soil water, i.e., the opposite of what is found in this study.

Another possibility is that fractionation processes occur during water extraction. Meißner et al., (2014) reported that treating soil samples with HCl to remove carbonates prior to water extraction led to a cryogenically-extracted water $\delta^{18}O$ in agreement with that of input water, whereas the $\delta^{18}O$ of cryogenically-extracted water from carbonate-rich soil samples was depleted by about 1‰ compared to input water. On the other hand, they found no effect of carbonate content on hydrogen isotopes. They suggested that the $\delta^{18}O$ depletion of extracted water was caused by oxygen isotope exchanges between soil

water and carbonates during the extraction, a process that should be temperature-dependent. Meißner et al. (2014) did not specify their extraction temperature but we expect it to be > 60°C, i.e. close to our extraction temperature of 80°C, so that we could expect a carbonate-induced isotope effect of comparable magnitude. If the presence of carbonates in the C horizon were responsible for a $\delta^{18}O$ depletion of extracted water from the deep soil samples of about 1‰, this would mean that the "true" soil water in this horizon should be shifted by about +1‰. This would slightly modify the SW-excess values but would not

cancel the observed isotopic offset between soil water and xylem water. Therefore, although the results of Meißner et al. (2014) are very relevant to our study, they cannot explain the isotopic offset observed here.

      The water content of rocks was quite high (*ca*. 12% in volume) and highly enriched in both $^{18}O$ and $^{2}H$ compared to bulk soil water. It is not clear what causes this enrichment of rock water compared to the surrounding soil. If root water uptake were causing it, this would mean that isotopic fractionation processes during rock water absorption by the roots enrich rock

water and deplete the water taken up by the plants. Without root water uptake, the isotope composition of rock water should then be similar to the surrounding soil, and root water uptake could thus explain a depletion of plant-accessed water compared to soil water, at least when rock water contributes significantly to the plant water use. However, the resulting isotopic offset should also vary over time and space because it would be related to the soil water and rock contents at the different depths. In contrast, we measured a rather constant offset in the dual-isotope space, driven mainly by $\delta^{2}H$ and poorly explained by

environmental variables. Similar results in the literature (although scarcely discussed) can be found at sites with contrasting plant and soil types, soil moisture regimes and lithology (Brooks et al., 2010; De Deurwaerder et al., 2018; Geris et al., 2017; Vargas et al., 2017; Zhao et al., 2016). Thus, although empirical evidence for an isotope separation between bulk and plant-accessed soil water pools is growing, our results do not support that this would be produced by isotopic fractionation during rock water absorption by the plants. Otherwise, we would expect both hydrogen and oxygen isotopes to be affected and the

isotope separation between plant and bulk soil waters to be weaker when soil water content is large. Instead, we found for all our trees a significant $\delta^{2}H$ offset between xylem and soil water sources (Fig. 5), even at times when soil water content was high (Fig. 1). Similarly, we do not think that branch evaporation is responsible for the reported isotopic offset (Martín-Gómez et al., 2017). If it were the case, we would expect the magnitude of the offset to vary over the season with evaporative demand and to affect both hydrogen and oxygen isotopes, i.e., the opposite of what we report here.

The SW-excess of xylem water collected from coarse roots was significantly more depleted than that from twigs (Fig. 4). Previous studies have shown that water in coarse or tap roots can exhibit significantly lower $\delta^{2}H$ relative to source water pools, for example in *Populus euphratica* (Zhao et al., 2016) and *Prosopis velutina* (Ellsworth and Williams, 2007). Moreover,

the $\delta^2H$ offsets between soil and root water observed in the former studies were of the same order of magnitude (*ca.* -20‰ for *P. euphratica* and ca. -7‰ for *P. velutina*) as observed here for *F. sylvatica* and *Q. robur* (Fig. 4 and Fig. 5). Interestingly, Zhao *et al.* (2016) were able to access plant water pools under positive pressure (using a syringe installed on the tree trunk) and found that this pool of water had the same isotopic composition as groundwater. They concluded that the depletion of bulk xylem water compared to soil water was caused by isotopic heterogeneity in the xylem water pools. In this context, Ellsworth & Williams (2007) attributed the $\delta^2H$ depletion in halophytic plants to their strong dependence on the symplastic pathway of water movement into the root. Symplastic transport of water is not only important during root water uptake, but also plays a relevant role in the storage dynamics of wood water, notably during periods of increasing tension in the xylem (Pfautsch et al., 2015). This pathway uses ray parenchyma cells and is used to maintain a two-way exchange of water between the phloem and the xylem. It is thus plausible that in roots, containing a higher fraction of parenchyma cells than stems (Morris et al., 2016), a relatively higher volume of water moving through the symplast could cause a strong depletion of bulk wood water, which is the water sampled during cryogenic extraction. Interestingly, ray and axial parenchyma can account for around 31% of total xylem tissue volume in both *F. sylvatica* and *Q. robur* (Morris et al., 2016) whilst storage water in the stem can account for up to 16% of daily transpiration in *F. sylvatica* (Köcher et al., 2013), and contribute even more in some subtropical tree species (Oliva Carrasco et al., 2015). Thus future studies are now required to explore the role of symplastic water transport and storage as a potential mechanism leading to the depletion of bulk wood water $\delta^2H$ compared to the actual source water signal. This mechanism may be quantitatively relevant for interpreting the isotopic composition of bulk xylem water in terms of source water and explaining the variability in SW-excess reported here.

Yet another explanation for the observed isotopic mismatch could be the development of non-monotonic isotopic profiles in the soil. Even if evaporative enrichment caused the top soil layers to be usually more enriched in both isotopes than deep soil layers (Fig. 2), the isotopic composition of intermediate layers may not always lay in-between those of the top and deep soils. Indeed, the water isotopic profile is usually disrupted after rains, whereby surface soil layers can become more depleted than deep soil (see May 23th and July 4th in Fig. 2). Following such events, the surface soil layer becomes enriched again within a few days depending on the evaporative demand but percolated rain water to intermediate depth may have created a soil layer between the deep and shallow soil with a more depleted isotopic composition. Because we only sampled surface (0-10cm) and deep soil (>70cm), it is possible that, during such event, the un-sampled intermediate soil layers holds water of similar isotopic than that of xylem water. In fact, a missing water source has been sometimes claimed to explain isotopic offsets between source and xylem water, even in studies when soil water sampling was performed every 10 cm (Bowling et al., 2017). We have several reasons to think that our sampling strategy is not what causes the isotopic separation between xylem and soil water. First, xylem $\delta^{18}O$ was always in the range of soil water $\delta^{18}O$ (Fig. 3). Non-monotonic soil isotopic profiles would imply that both soil water $\delta^{18}O$ and $\delta^2H$ are more depleted in a certain layer, as they should be proportional (Barbecot et al., 2018). Thus, if root water uptake was sourced in a soil layer with relatively depleted soil water, xylem water should be depleted in both isotopes, not just in $\delta^2H$. Second, non-monotonic soil isotopic profiles may occur under some circumstances, that can be

identified with the dates when soil $\delta^{18}O$ and $\delta^2H$ data does not spread on a clearly identified line in the dual isotope space (Table S3). Yet, the estimated SW-excess was negative for both species in all sampling campaigns, i.e., following rain events and subsequent soil water percolation and isotopic mixing, short drought periods enhancing surface evaporative enrichment, as well as periods with a likely different depth of root water uptake. Third, it is very unlikely that depleted winter precipitation can be stored in this intermediate soil layers and that this would be behind the observed depleted xylem water. This is because, again, winter precipitation should be reflected in both isotopes and also, but also because of the coarse soil texture of the intermediate soil layer (Table S1), which should lead to fast infiltration rates of winter precipitation to the sampled deep soil.

### 4.2. Estimation of source contribution to xylem water

Based on results from the Bayesian isotope mixing model with uncorrected $\delta^{18}O$ and $\delta^2H$ signals, we found that both *F. sylvatica* and *Q. robur* used a mixture of both top and deep soil water throughout the 2017 growing season, with marginal contributions of streamwater (or groundwater) (Fig. 6). The relative contribution of these water sources was also quite dynamic, leading for example to rapid shifts towards the deep soil in late summer when soil was progressively drying. According to this analysis, we also found that *Q. robur* used more deep soil water than *F. sylvatica*, which is consistent with the idea that oaks tend to invest more in deep roots (Rosengren et al., 2005), whilst beech trees usually have a dense fine root network in top soil layers (Leuschner et al., 2001). When using $\delta^{18}O$ alone or in combination with SW-excess-corrected $\delta^2H$ signals, the contribution of stream/groundwater to the water sources of *F. sylvatica* trees increased significantly (Fig. 7). Typically, soils in this region are acidic and sandy and contain a hardpan that restricts access to the groundwater. In contrast, the riparian forest in the present study was located on a slightly more basic soil (Table S1) with a rocky layer of limestone that provided a water connection to the groundwater. This may be a reason why *F. sylvatica* is consistently restricted to a few particular sites in its southern range (Lafontaine et al., 2014; Timbal and Ducousso, 2010).

Studies applying isotope mixing models such as *MixSIAR* to study plant water sources usually assume no isotopic offset between xylem water and source water (Barbeta et al., 2015; Evaristo et al., 2017; Palacio et al., 2014; Rothfuss and Javaux, 2017). However, we have shown evidence that $\delta^2H$ offsets are present as a result of isotopic fractionation processes either within soil water pools or within plant tissues, and made an attempt to evaluate their effects on the estimates of plant water sources by comparing mixing models with different input data. This exercise was made with the purpose of providing a sensitivity analysis. However, its use in other sites and plant species should be made with caution, as it is very likely that the observed $\delta^2H$ offset may display different patterns depending on other water sources. By correcting xylem $\delta^2H$ based on the SW-excess, we found that the contribution of stream water had been underestimated compared with the classical approach (Fig. 7). Contrary to recent studies that reported a low sensitivity of Bayesian isotope mixing models to $\delta^2H$ offsets (Evaristo et al., 2017), we found that the plant water source estimations also varied between models using either $\delta^{18}O$ only, $\delta^2H$ only or models using both isotopes. In addition, this disparity caused by $\delta^2H$ offsets cannot be solved by using only the apparently

non-fractionating $\delta^{18}O$ as having one isotopic tracer is insufficient to distinguish between water sources in many cases (Fig. 8).

Bayesian mixing models were shown to perform best in a recent comparison of approaches to quantify root water uptake (Rothfuss and Javaux, 2017). These comparisons were based however on the assumption that no fractionation occurs during root water uptake and redistribution within the plant tissues. Therefore, the application of these models may not be suitable in studies where $\delta^2H$ offsets are suspected. Interestingly, we found that correcting the xylem $\delta^2H$ data using the SW-excess gave stronger correlations with environmental data and allowed for a more parsimonious interpretation, such as an increase in top soil water uptake with cumulative 5-day rainfall amounts (Table 1). Based on these correlations, such a correction using the SW-excess appeared to improve the predictive power of the dual-isotope approach. However, systematically correcting xylem data with the SW-excess may also be in cases where other water sources, not belonging to the soil water line, contribute to the xylem isotopic signal. Finally, spatiotemporal dynamics in the soil water isotope profile could also complicate the concept of the soil water isotope line and thus the SW-excess. The fact that we sometimes observed positive SW-excess indicates that we do not only correct for one single fractionation factor, and demonstrates the limitation of the SW-excess correction proposed here. In addition, in our study, the correlation of soil water $\delta^{18}O$ and $\delta^2H$ was not always significant (Table S3), although this did not seem to affect the SW-excess quantitatively. However, the concept of the SW-excess becomes less meaningful when soil water isotopes are not significantly correlated, since the error associated to the regression coefficients could be of the same magnitude than that of the calculated SW-excess.

## 5. Conclusion

In the light of our results and other recent studies, either conducted under controlled conditions (Oerter et al., 2014; Vargas et al., 2017) or in the field (Evaristo et al., 2017; Oerter and Bowen, 2017; Oshun et al., 2015; Zhao et al., 2016), evidence for fractionation processes occurring at the soil-root interface or within plant woody tissues is growing. These processes may complicate or prevent the identification of plant water sources, especially when they remain unnoticed. Importantly, $\delta^2H$ fractionation during or after root water uptake seems to extend beyond plants growing in salty and dry environments (Ellsworth and Williams, 2007). This should now motivate researchers to develop hypothesis-driven studies focused on two main lines. Firstly, to couple the study of physicochemical fractionation processes in the unsaturated zone and their repercussions on plant absorbed water, covering a range of soil properties and water content (as illustrated by Vargas *et al.*, 2017). Secondly, to obtain a better understanding of the isotopic dynamics of water pools within plant tissues, notably those associated with plant storage water and its dynamics (Pfautsch et al., 2015) which will require developing new extraction methods for xylem water (*e.g.* a method allowing to separate younger xylem vessel water from older water stored outside the xylem, see Zhao et al., (2016)). Interestingly, if there are isotopically distinct water pools within plant stems, this could be used to quantify the contribution and age of wood water storage. On the other hand, this would clearly complicate the use of Bayesian isotope mixing models to partition plant water sources. Nonetheless a better understanding of what causes this isotopic separation between xylem and

source water is urgently needed to constrain their respective contributions in isotope mixing models, as already applied in other applications such as in food web studies (Phillips et al., 2014), and provide more parsimonious plant water source estimations.

Although the present dataset does not allow us to assess definitively which are the ecohydrological mechanisms that have assisted the persistence of *F. sylvatica* in this riparian forest for at least the last 40,000 years, we can rule out a dominance of water uptake from the stream, as found in other riparian tree species (Bowling et al., 2017; Dawson and Ehleringer, 1991; Oerter et al., 2019). We cannot rule out that the contribution of stream or ground water, although small, is responsible for the long-term persistence of *F. sylvatica* in this area, but the presence of a layer of weathered limestone seems also to confer advantageous soil characteristics to *F. sylvatica*. Indeed, the C horizon is located within the rooting depth of both studied species (at a depth 50-120 cm) and has higher clay and finer sand fractions than the A and B horizons, and than soils in the surrounding region. Our findings that this (deep soil) C horizon seems to contribute significantly to the water source of beech trees indicates that the higher water holding capacity of soils around the Ciron river could be responsible for the long-term persistence of *F. sylvatica* in this valley. It is worth noting that, although 2017 was characterised by relatively low soil water content early in the season, the precipitation input was evenly spread over the growing season and the atmospheric evaporative demand did not stay high for prolonged periods (Fig 1). Therefore, it would be interesting to perform a similar survey during a year with a prolonged drought and high evaporative demand (e.g. in the top percentile hotter and dryer years of the last 25 years) to observe how the contribution of ground or river water and deep soil water varied in these years. This would be necessary in order to perform projections of the persistence likelihood of *F. sylvatica* in this valley exposed to a potentially drier climate in the future.

*Data availability*

The data collected in this study is available upon request to the authors.

*Author contributions*

A.B., J.O. and L.W. designed the study. A.B., L.C., B.F., S.J., S.W. and T.E.G. conducted the field work. S.J., S.W, J.O. and L.W. developed and tested the cryogenic water extraction vacuum line, and produced the protocols and code to process water isotope data. B.F. and L.C. ran all the samples through the water extraction line. A.B., L.C., B.F. and S.J., conducted the water isotope analyses. A.B. and J.O. analysed the data and wrote a first draft of the manuscript. All the other authors edited the manuscript in several rounds of revision.

*Competing interests*

The authors declare that they have not conflict of interest.

*Acknowledgements*

This study has been carried out on the HydroBeech, ClimBeech and MicroMic projects with financial support from the French National Research Agency (ANR) in the frame of the Investments for the future Programme, within the Cluster of Excellence COTE (ANR-10-LABX-45). A.B. acknowledges an IdEx Bordeaux postdoctoral fellowship from the Université de Bordeaux (contract no. 22001162). This project has also received funding from the European Research Council (ERC) under the European Union's Seventh Framework Program (FP7/2007-2013) (grant agreement no. 338264) awarded to L.W., a Marie Skłodowska-Curie Intra-European fellowship (Grant Agreement No. 653223) awarded to T.E.G., the French Agence National de la Recherche (ANR) (grant agreement no. ANR-13-BS06-0005-01) awarded to J.O. and the Aquitaine Region project Athene awarded to L.W (2016-1R20301-00007218).

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

**Table 1: Output of the generalized linear mixed models computed with the source contributions estimated with different input data and environmental independent variables. Dual implies the use of both water isotopes. Individual models are run per each plant-water source and input data type. For each model effect, the β estimate (standardized correlation coefficient) is shown. Marginal $R^2$ corresponds to the variance in source contribution explained by the model independent variables. Significant effects are highlighted in bold and with asterisks (\*$P$<0.05, \*\*$P$<0.01, \*\*\*$P$<0.001).**


| Input data | Source | Rainfall (5-day amount) | VPD (5-day average) | Top soil moisture | Deep soil moisture | Marginal $R^2$ |
|---|---|---|---|---|---|---|
| Dual | Top soil | **0.428\*** | 0.004 | -0.002 | -0.081 | 0.140 |
| | Deep soil | -0.531 | -0.048 | -0.092 | 0.128 | 0.232 |
| | Stream water | 0.138 | 0.138 | 0.298 | -0.111 | 0.095 |
| Dual, $\delta^2$H corrected | Top soil | **0.425\*\*** | 0.033 | 0.035 | **-0.664\*\*\*** | 0.336 |
| | Deep soil | **-0.422\*** | 0.013 | -0.221 | **0.641\*\*\*** | 0.393 |
| | Stream water | 0.014 | -0.028 | 0.415\* | -0.258 | 0.203 |
| Only $\delta^{18}$O | Top soil | -0.153 | **-0.429\*** | 0.271 | **0.372\*** | 0.272 |
| | Deep soil | 0.158 | 0.277 | **-0.608\*\*\*** | **-0.320\*** | 0.398 |
| | Stream water | 0.128 | **0.474\*** | -0.168 | -0.203 | 0.227 |
| Only $\delta^2$H | Top soil | 0.375 | 0.043 | -0.065 | 0.101 | 0.154 |
| | Deep soil | -0.021 | 0.350 | -0.049 | -0.158 | 0.160 |
| | Stream water | **-0.682\*\*** | **-0.661\*\*\*** | 0.215 | **0.316\*** | 0.526 |

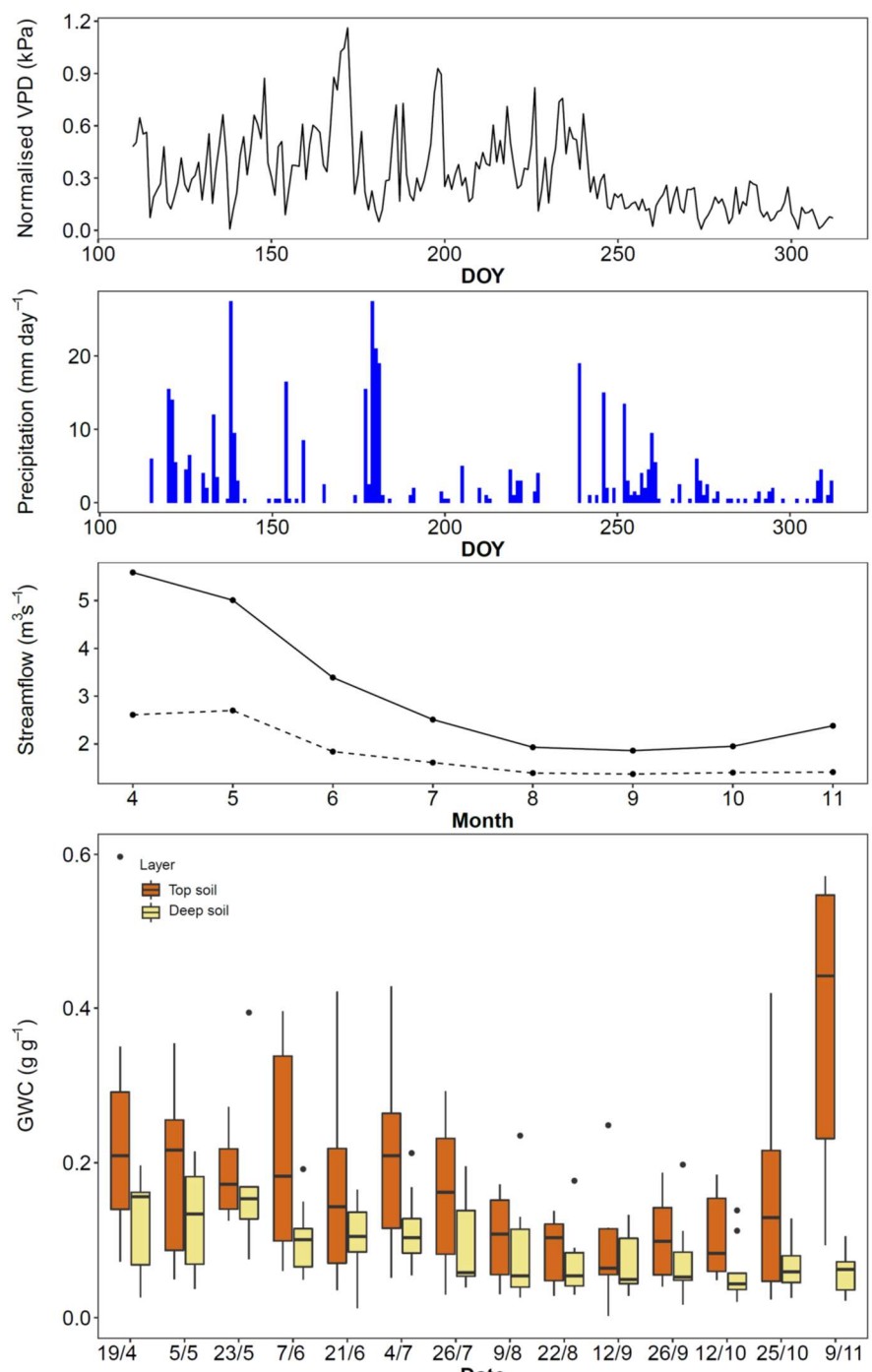

**Figure 1: Environmental conditions in the riparian forest along the Ciron river during the 2017 growing season. From the top panel to the bottom; daily vapour pressure deficit (VPD), daily precipitation, monthly streamflow for 2017 (dashed line) and the 2010-2017 period (solid line) and gravimetric water content of top soil (0-10 cm) and deep soil (ranging from 50 to 120 cm).**

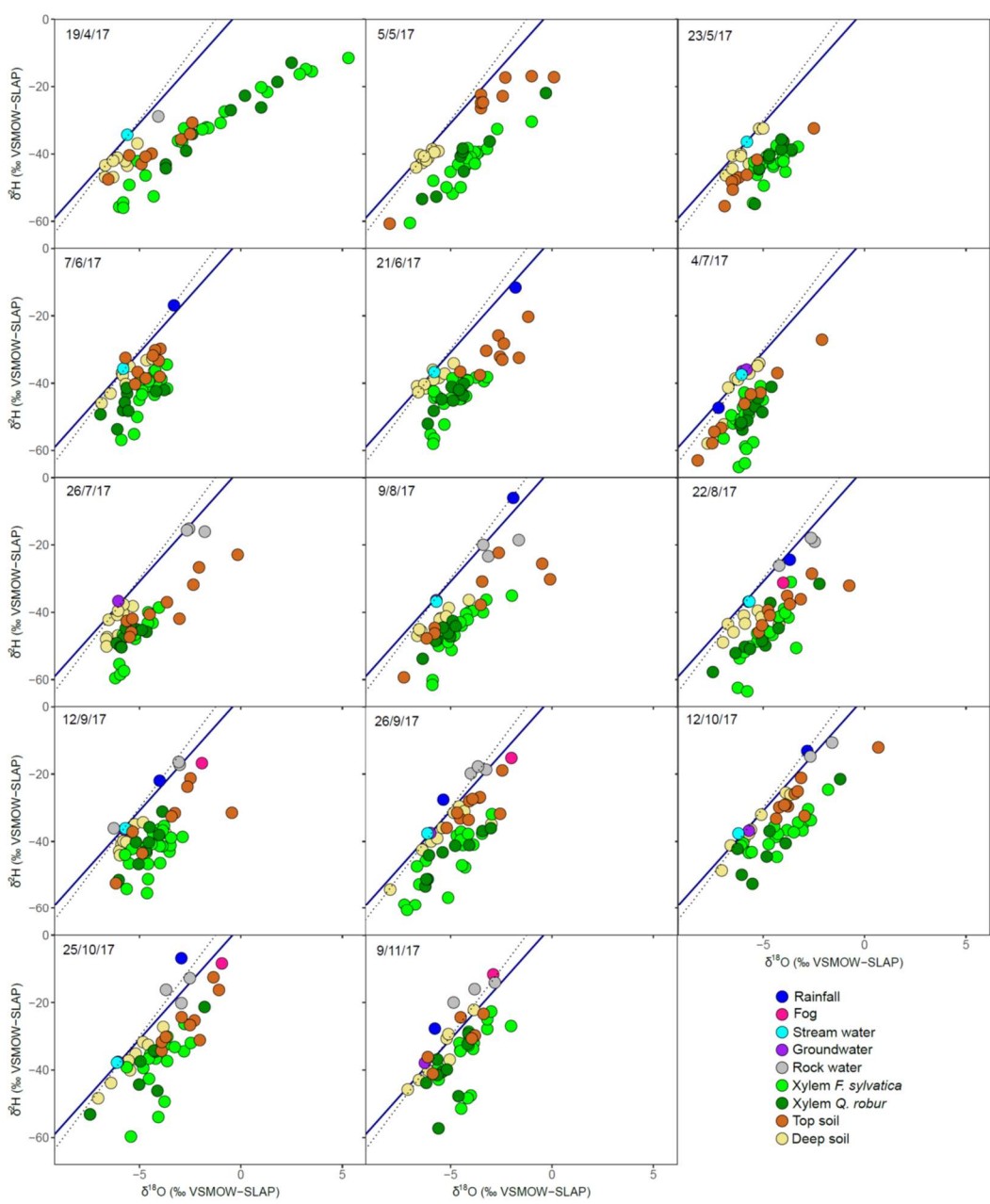


**Figure 2: Dual isotope (δ²H and δ¹⁸O) plot of xylem water of the two studied species (*F. sylvatica* and *Q. robur*) and its potential sources (soil water at two depths, groundwater, stream, rain, fog and rock water) for every sampling campaign conducted in 2017. The blue line indicates the LMWL whereas the dashed black line indicates the GMWL.**

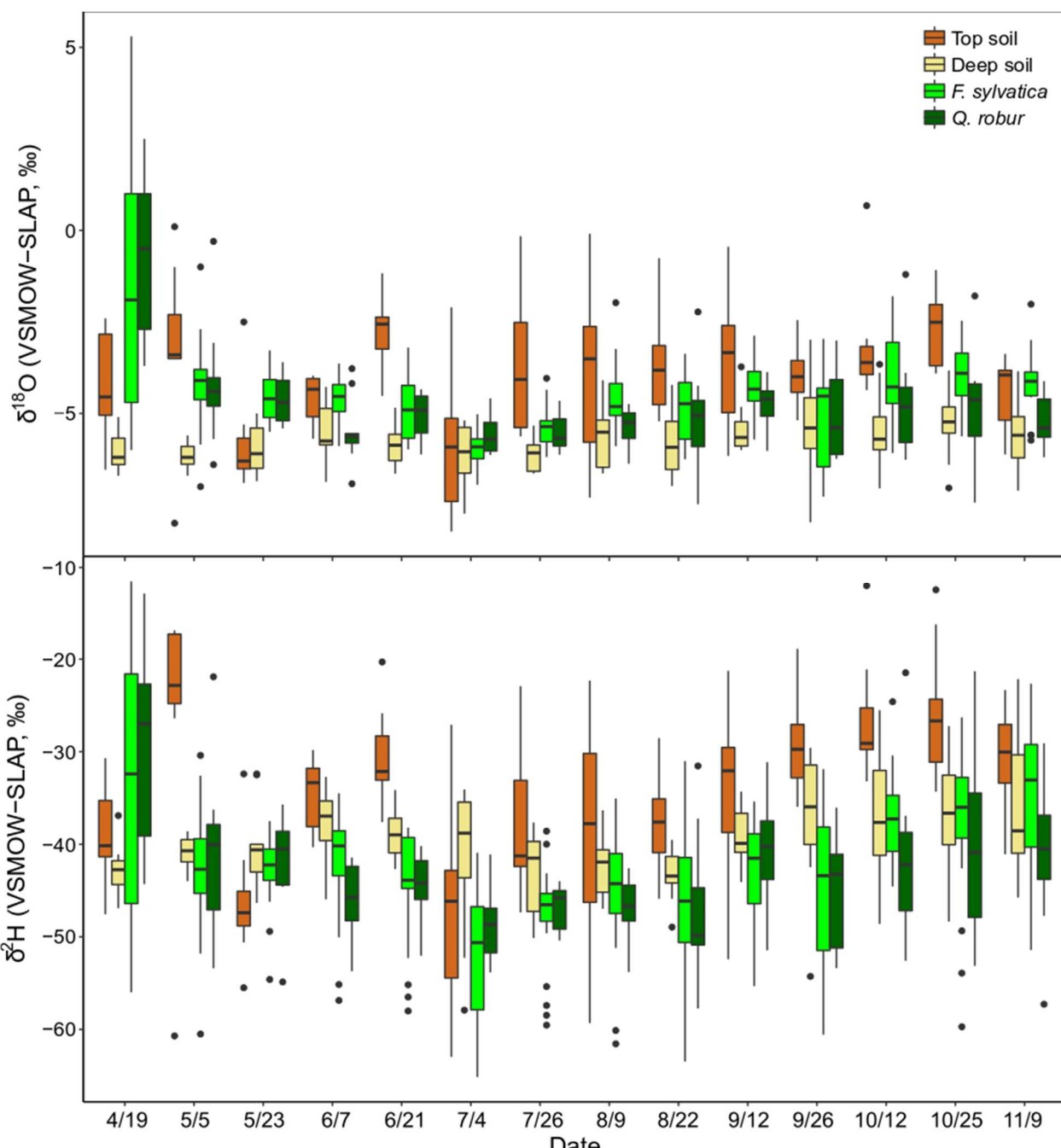


**Figure 3: Isotopic composition of top soil water (0-10 cm), deep soil water (50-120) and xylem water of the two studied species (*F. sylvatica* and *Q. robur*) for each sampling campaign. Data is pooled over the three studied plots. Box size represents the interquartile range, the black line is the median, the whiskers indicate variability outside the upper and lower quartiles, and individual points are outliers.**



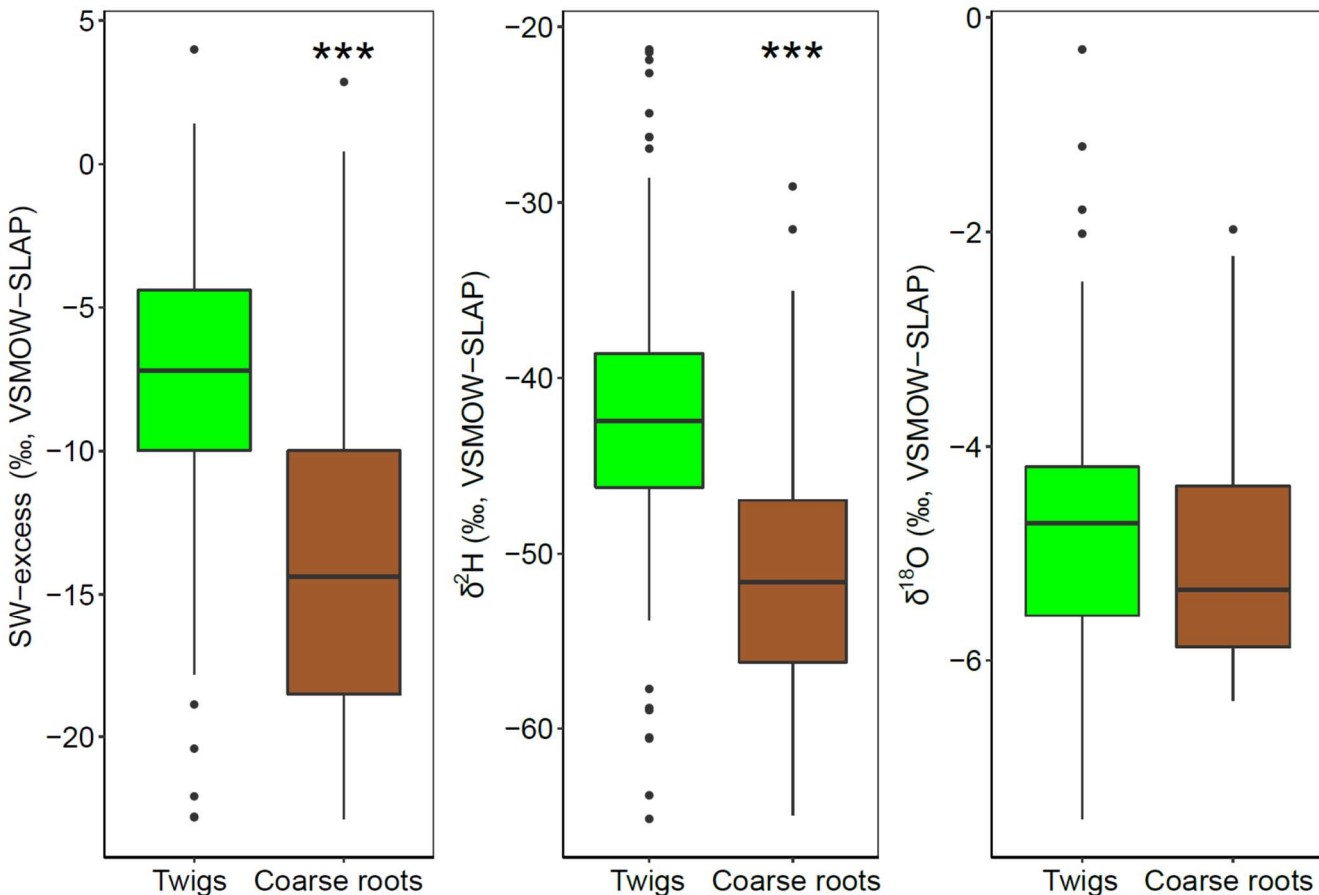

**Figure 4: SW-excess, $\delta^2$H and $\delta^{18}$O of xylem samples of dominant *F. sylvatica* and *Q. robur* depending on the part of the tree sample**
**(twigs or coarse roots). Significant differences between twigs and coarse roots (*P*<0.001) are highlighted with asterisks (\*\*\*). Box size represents the interquartile range, the black line is the median, the whiskers indicate variability outside the upper and lower quartiles, and individual points are outliers. Xylem samples from the first sampling campaign were excluded from the analysis because of probable winter branch evaporation (see text).**


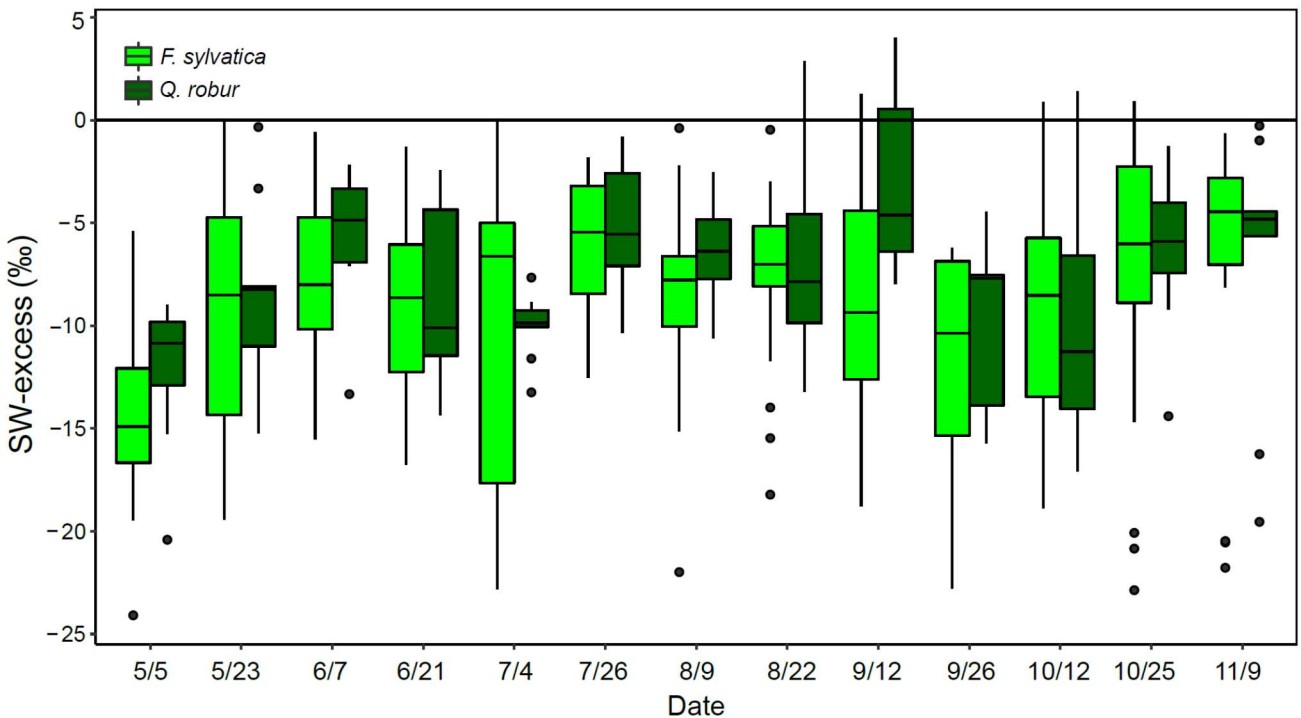

**Figure 5: Temporal variations of SW-excess of *F. sylvatica* and *Q. robur* xylem samples over the 2017 growing season. Box size represents the interquartile range, the black line is the median, the whiskers indicate variability outside the upper and lower quartiles, and individual points are outliers.**



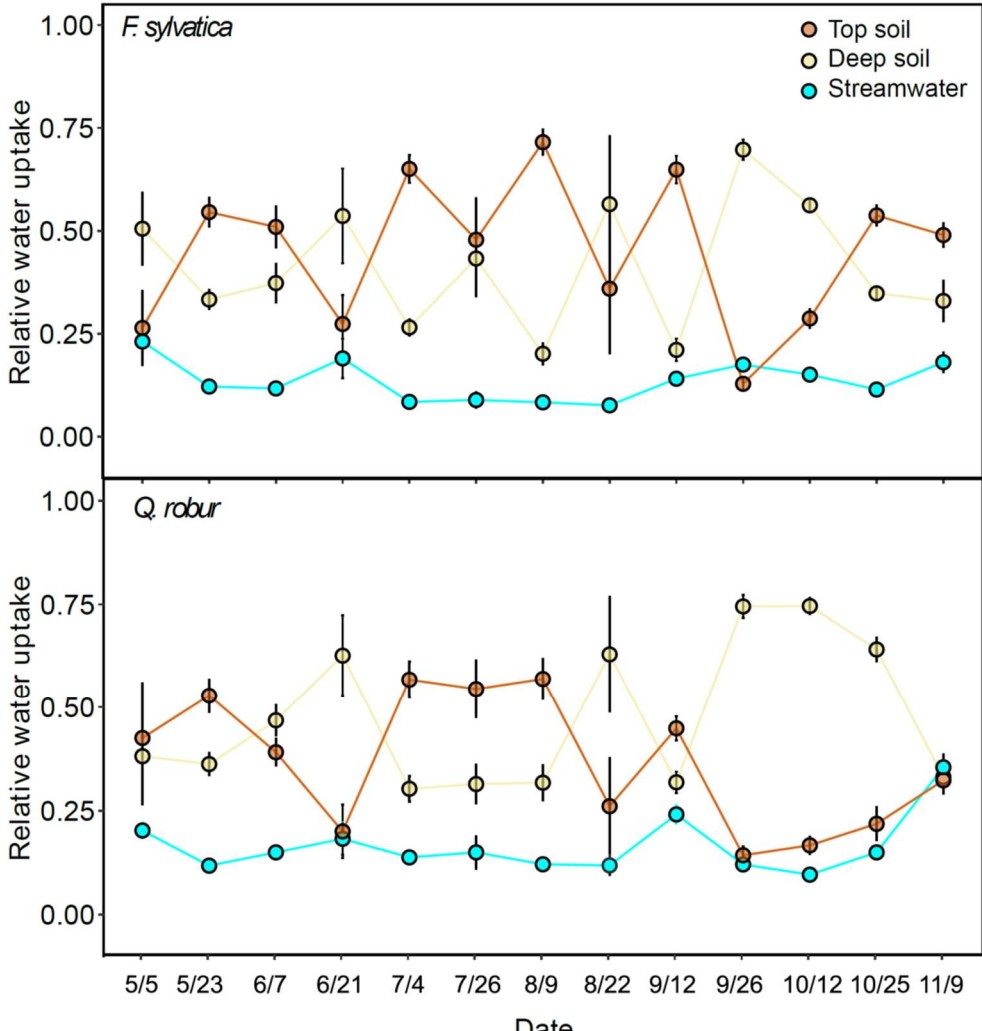

**Figure 6: Relative water uptake from each of the plant water sources considered: topsoil, deep soil and stream water (indistinguishable with groundwater), as estimated with *MixSIAR* for the dominant trees of *F. sylvatica* (top panel) and *Q. robur* (bottom panel). The error bars correspond to the standard deviation (N=15 for *F. sylvatica* and N=9 for *Q. robur*). These proportions were estimated with uncorrected δ²H and δ¹⁸O values for xylem water.**

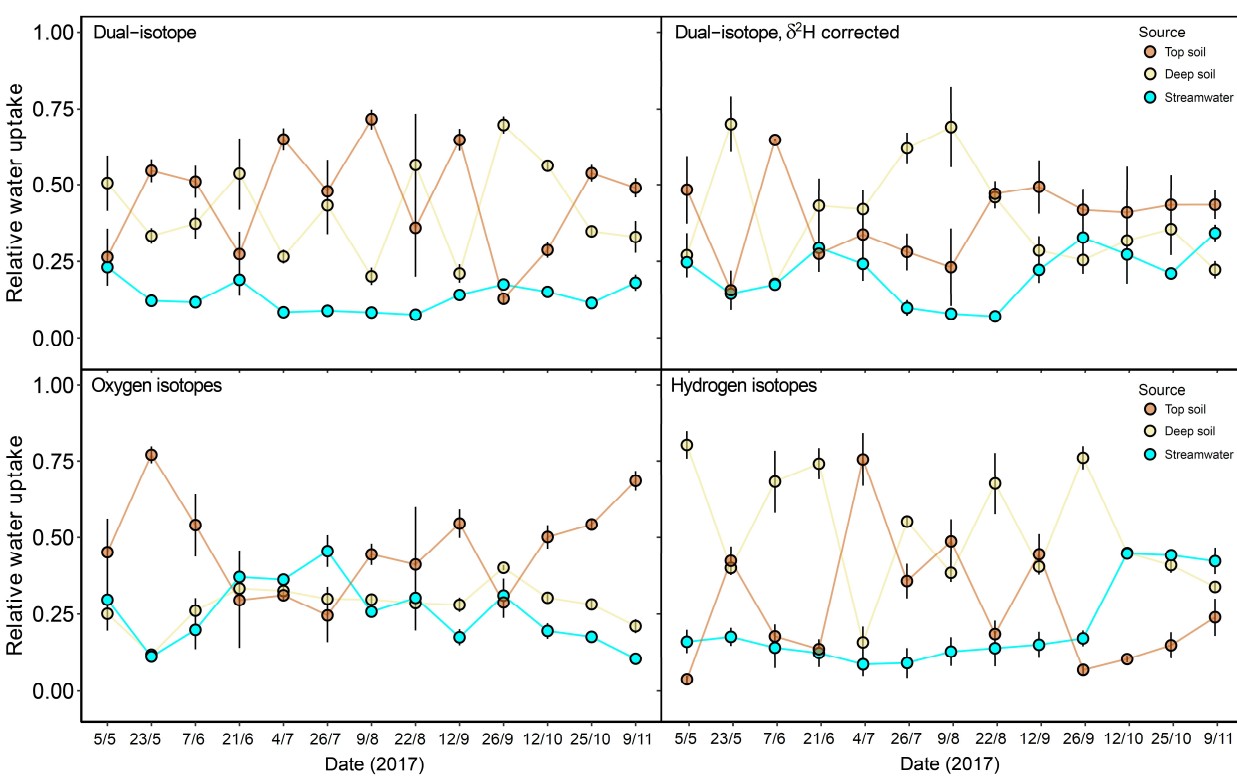


**Figure 7: Relative water uptake from each of the plant water sources considered: topsoil, deep soil and stream water (indistinguishable with groundwater), as estimated with *MixSIAR* for the dominant trees of *F. sylvatica*. The input data is different for each of the four panel, top left; uncorrected $\delta^2$H and $\delta^{18}$O, top right; $\delta^2$H (corrected for SW-excess) and $\delta^{18}$O, bottom right; only $\delta^{18}$O and bottom right; only $\delta^2$H. The error bars represent standard deviations (N=3).**




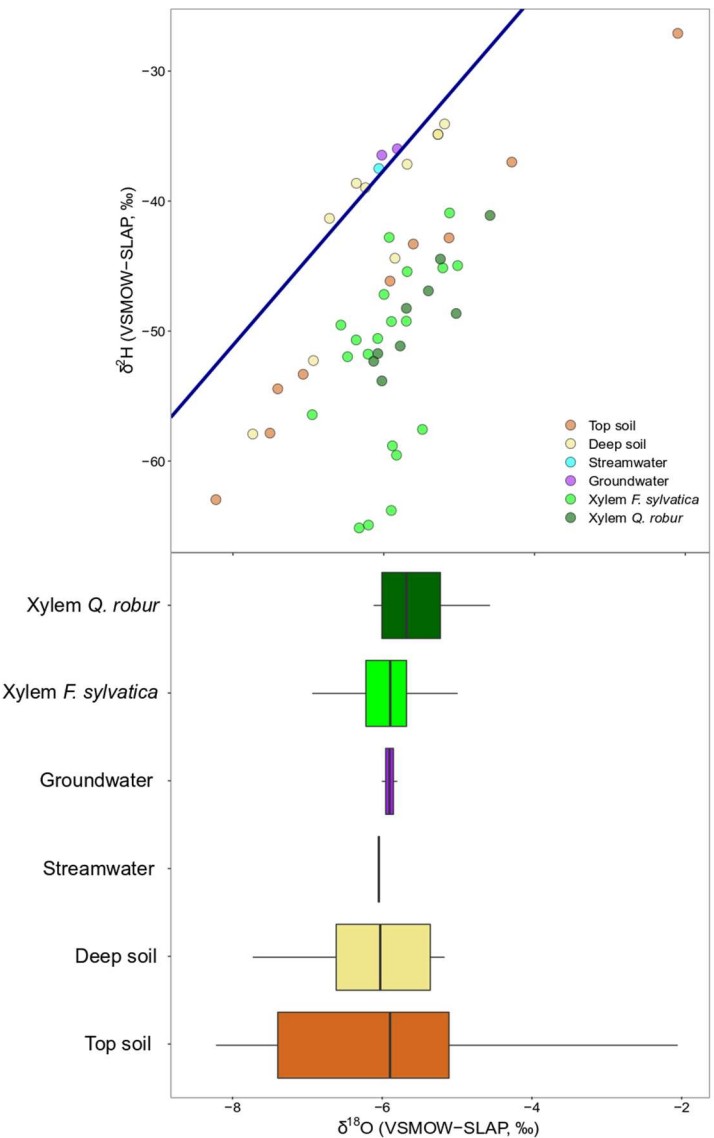

**Figure 8: Differences in the isotopic composition of xylem water and its potential sources using one isotope. The top panel depicts the dual-isotope plot for a single date (July 4th), with xylem water and sources. The bottom panel is the boxplot of the $\delta^{18}O$ values for xylem water of *F. sylvatica* and *Q. robur* and each of the potential sources. Box size represents the interquartile range, the black line is the median, the whiskers indicate variability outside the upper and lower quartiles, and individual points are outliers.**


