# Peer review of "Unexplained hydrogen isotope offsets complicate the identification and quantification of tree water sources in a riparian forest"

_Hydrology and Earth System Sciences, 2018_

## Referee Comment (RC1) · Song (Referee) · 1 Feb 2019

In this manuscript, Barbeta et al. presents a field-based water isotope dataset that was collected for the purpose of investigating potential water sources of two broad-leaved tree species in a temperate, riparian forest. One of the main findings of their study was that hydrogen isotope ratios in the xylem water of the two studied tree species were generally more depleted than those of soil water at different depths and other potential water sources. This suggests potential fractionation effects associated with hydrogen isotopes at the soil-root interface and/or within plant tissues. This is the second time I am reviewing this paper. The authors have made great efforts addressing the concerns

that were raised in my previous reveiw report, which I appreciate. I think the ms in its current form is strong and I would thus recommend it for publication in HESS.

---

## Referee Comment (RC2) · Juan Pedro Ferrio Diaz (Referee) · 12 Feb 2019

General comments

The work by Barbeta et al. highlights one critical issue in the application of stable isotopes as hydrological tracers for the study of plant water uptake patterns. Methods are described properly, but some clarifications are needed. In general, the data is clearly presented and the discussion is well written and focused. Some improvements could be made in the figures in order to make them more self-explanatory. Overall, the manuscript is timely, shows good quality data and makes a significant contribution to ecohydrology.

Specific comments

The calculation of the SW-excess, instead of the deviation from the LMWL, is properly justified in the methods section, and appears as a reasonable alternative for the context of this study. However, I am concerned about the fact that soil data did not always show a single evaporative line (e.g. 5/5/17; 23/5/17; 4/7/17). It would be useful to show the fitting statistics for these regressions (e.g. r2, intercept, slope, p-value). One way to consider this uncertainty is to take into account the confidence intervals of this slope in order to recalculate the errors associated to the SW-excess, and eventually include this as a kind of "analytical error" term in the models.

Regarding the degree of mixing between precipitation and different soil water pools (e.g. line 59, lines 70-75, lines 380-381), and the effect of recent rainfall on soil water d18O and d2H (lines 287-290), it is particularly suitable the discussion about rewetting-drying cycles presented in (Tang and Feng 2001). Indeed, (Tang and Feng 2001) also found little effect of recent precipitation below 50 cm depth, with the exception of particularly strong rain events.

Figure 4. The point that coarse roots show larger fractionation than twigs does not support the "fractionation during water uptake" hypothesis, but favours the option of some kind of isotopic-exchange undergoing in stored water. During a previous field study (Martín-Gómez et al. 2017), we did some preliminary tests comparing twig water with water extracted from trunk cores. Interestingly, and in line with the present study, we found a depletion in d2H of trunk water of about 10‰ as compared to soil and twig samples, although this was not consistent across tree species and sampling times (Martín-Gómez et al., unpublished). The apparent "bypass" of the root fractionation along the path from soil to twigs described by Barbeta et al., and the differences between xylem sap and distilled xylem water shown by Zhao et al. (2016), suggest that fractionation processes associated with water storage could be the key for the observed changes, and certainly deserve further studies.

Technical comments

Figure 1. Although the measurements were taken at different time intervals, it would be desirable to adjust all the panels to a single scale in the X axis.

In Figures 2, 6, 7, 8a, to facilitate interpretation, I would combine fill colours with different symbol shape, and keep them unified throughout the manuscript. For example, circles could represent xylem water, squares soil water, diamonds for fog and rainfall, upward triangles for stream water, ground water and rock water.

Line 165. If I understood well, 3 of the beech trees and 1 of the oaks were sampled from the roots, whereas in the rest of the trees the sampling was based on twigs. According to Figure 4, the observed d2H-depletion was much stronger in coarse roots than in twigs. However, I wonder whether the significant test shown simply indicates that twigs and roots have different SW-excess, or it shows the significance of the SW-excess (i.e. divergence from SW-excess=0). In this regard, the text citing the figure does not clarify this point: "we found differences in SW-excess when the xylem water was collected from coarse roots rather than from twigs (Fig. 4)." In any case, since SW-excess in roots is about twice that found in the twigs, it is worth to indicate them separately in the rest of the graphs.

References

Martín-Gómez P, Aguilera M, Pemán J, et al (2017) Contrasting ecophysiological strategies related to drought: the case of a mixed stand of Scots pine (Pinus sylvestris) and a submediterranean oak (Quercus subpyrenaica). Tree Physiol 37:1478–1492. doi: 10.1093/treephys/tpx101

Tang KL, Feng XH (2001) The effect of soil hydrology on the oxygen and hydrogen isotopic compositions of plants' source water. Earth Planet Sci Lett 185:355–367. doi: 10.1016/S0012-821X(00)00385-X

Zhao L, Wang L, Cernusak LA, et al (2016) Significant Difference in Hydrogen Isotope Composition Between Xylem and Tissue Water in Populus Euphratica. Plant Cell Environ 39:1848–1857. doi: 10.1111/pce.12753

---

## Author Comment (AC1) · 28 Feb 2019

We appreciate the positive assessment of the referee as well as his suggestions to improve the manuscript made in the previous round of review. In order to acknowledge his contribution, the interactive discussion of this previous round of review can be found in the manuscript files of the following Discussion paper, that was initally rejected:

Barbeta, A., Jones, S. P., Clavé, L., Wingate, L., Gimeno, T. E., Fréjaville, B., Wohl, S., and Ogée, J.: Hydrogen isotope fractionation affects the identification and quantification of tree water sources in a riparian forest, Hydrol. Earth Syst. Sci. Discuss., https://doi.org/10.5194/hess-2018-402, 2018.

---

## Author Comment (AC2) · 28 Feb 2019

**Authors' responses in bold.**

General comments

The work by Barbeta et al. highlights one critical issue in the application of stable isotopes as hydrological tracers for the study of plant water uptake patterns. Methods are described properly, but some clarifications are needed. In general, the data is clearly presented and the discussion is well written and focused. Some improvements could be made in the figures in order to make them more self-explanatory. Overall, the manuscript is timely, shows good quality data and makes a significant contribution to ecohydrology.

**We appreciate the positive assessment of the reviewer. Following his comments, we have amended the manuscript to improve its clarity and added additional analyses that were not shown previously. We agree with the reviewer that those analyses are helpful for the understanding of the data presented (notably, for the SW-excess).**

Specific comments

The calculation of the SW-excess, instead of the deviation from the LMWL, is properly justified in the methods section, and appears as a reasonable alternative for the context of this study. However, I am concerned about the fact that soil data did not always show a single evaporative line (e.g. 5/5/17; 23/5/17; 4/7/17). It would be useful to show the fitting statistics for these regressions (e.g. r2, intercept, slope, p-value). One way to consider this uncertainty is to take into account the confidence intervals of this slope in order to recalculate the errors associated to the SW-excess, and eventually include this as a kind of "analytical error" term in the models.

**We agree with the reviewer that the fitting of the soil water line may influence the calculation of the SW-excess. Importantly, non-significant correlations of the different soil water samples from the same sampling date and plot can reduce the relevance of the SW-excess. In the revised version of the manuscript, we now present a new table with the r-squared, intercept, slope and *p*-value of the soil water lines for each plot and sampling date (Table S3). In addition, we re-ran the analyses without those values corresponding to cases in which the soil water line regression was not significant. However, those cases did not have a significantly different SW-excess compared to cases with significant soil water line regressions. Accordingly, with have now added the following text:**

**In Results:**

**"The linear regression of the soil water line was significant for most of the sampling dates and plots, as shown in Table S3. Consequently, we removed from the multivariate analysis of the SW-excess those data corresponding to sampling dates and plots that did not present a significant soil water line regressions. Still, the SW-excess did not significantly differ between cases with significant soil water lines and cases with non-significant soil water lines (P = 0.45)."**

**In Discussion:**

**"In addition, in our study, the correlation of soil water $\delta^{18}O$ and $\delta^{2}H$ was not always significant (Table S3), although this did not seem to affect the SW-excess quantitatively. However, the concept of the SW-**

*excess becomes less meaningful when soil water isotopes are not significantly correlated, since the error associated to the regression coefficients could be of the same magnitude than that of the calculated SW-excess"*

Regarding the degree of mixing between precipitation and different soil water pools (e.g. line 59, lines 70-75, lines 380-381), and the effect of recent rainfall on soil water d18O and d2H (lines 287-290), it is particularly suitable the discussion about rewetting drying cycles presented in (Tang and Feng 2001). Indeed, (Tang and Feng 2001) also found little effect of recent precipitation below 50 cm depth, with the exception of particularly strong rain events.

**Unfortunately, we did not know this study from Tang and Feng, that reports seasonal dynamics of the isotopic mixing of rain and soil water, depending on rain amount and soil water status. We have now included citation of this study in the fragments mentioned by the reviewer.**

Figure 4. The point that coarse roots show larger fractionation than twigs does not support the "fractionation during water uptake" hypothesis, but favours the option of some kind of isotopic-exchange undergoing in stored water. During a previous field study (Martín-Gómez et al. 2017), we did some preliminary tests comparing twig water with water extracted from trunk cores. Interestingly, and in line with the present study, we found a depletion in d2H of trunk water of about 10‰ as compared to soil and twig samples, although this was not consistent across tree species and sampling times (Martín-Gómez et al., unpublished). The apparent "bypass" of the root fractionation along the path from soil to twigs described by Barbeta et al., and the differences between xylem sap and distilled xylem water shown by Zhao et al. (2016), suggest that fractionation processes associated with water storage could be the key for the observed changes, and certainly deserve further studies.

**This preliminary data on the relative depletion in $\delta^2$H of water in trunk cores mentioned by the reviewer does indeed coincide in sign and magnitude with our results and those of Zhao et al. (2016). We thus agree with reviewer that the hypothesis of fractionation between plant-internal water pools has more support by our findings than fractionation occurring during root water uptake. We think that we clearly state this in the Discussion of this manuscript. Nevertheless, further experiments targeted to this mechanism are required to test this hypothesis.**

Technical comments

Figure 1. Although the measurements were taken at different time intervals, it would be desirable to adjust all the panels to a single scale in the X axis. In Figures 2, 6, 7, 8a, to facilitate interpretation, I would combine fill colours with different symbol shape, and keep them unified throughout the manuscript. For example, circles could represent xylem water, squares soil water, diamonds for fog and rainfall, upward triangles for stream water, ground water and rock water.

**We appreciate the constructive comments of the reviewer. This figure has already been modified during a previous round of review. In fact, the time axis in Figure 1 was originally presented as it is now suggested by the reviewer but was changed to its present form following the advice of a previous reviewer. We think that the current version (with separate time axes) is more correct and keeps a good readability. Regarding the dual isotope plots, given that nowadays a large portion of paper reads are online, the color palette used for the figures is clear enough to distinguish between the different groups depicted and adding different shapes would not increase substantially the clarity of the figures.**

Line 165. If I understood well, 3 of the beech trees and 1 of the oaks were sampled from the roots, whereas in the rest of the trees the sampling was based on twigs. According to Figure 4, the observed d2H-depletion was much stronger in coarse roots than in twigs. However, I wonder whether the significant test shown simply indicates that twigs and roots have different SW-excess, or it shows the significance of the SW-excess (i.e. divergence from SW-excess=0). In this regard, the text citing the figure does not clarify this point: "we found differences in SW-excess when the xylem water was collected from coarse roots rather than from twigs (Fig. 4)." In any case, since SW-excess in roots is about twice that found in the twigs, it is worth to indicate them separately in the rest of the graphs.

**We have now clarified this point in both the figure legend and the text. We have modified the sentence in Discussion mentioned by the reviewer (see below). We also specified in the legend of Figure 4 that the significance highlighted in the plot with an asterisk refers to the difference in the isotopic composition and SW-excess between the water extracted from coarse roots and twigs. We found more appropriated pooling the data of coarse roots and twigs in the other figures. Figure 4 already highlights this difference, and the messages provided by the other figures would be more difficult to interpret if we add this additional factor. However, we included the effect of this factor in the analysis of the SW-excess (Table S2).**

**In Discussion:**

*"The SW-excess of xylem water collected from coarse roots was significantly more depleted than when xylem water collected from twigs"*

References

Martín-Gómez P, Aguilera M, Pemán J, et al (2017) Contrasting ecophysiological strategies related to drought: the case of a mixed stand of Scots pine (Pinus sylvestris) and a submediterranean oak (Quercus subpyrenaica). Tree Physiol 37:1478–1492. doi: 10.1093/treephys/tpx101

Tang KL, Feng XH (2001) The effect of soil hydrology on the oxygen and hydrogen isotopic compositions of plants' source water. Earth Planet Sci Lett 185:355–367. doi: 10.1016/S0012-821X(00)00385-X

Zhao L, Wang L, Cernusak LA, et al (2016) Significant Difference in Hydrogen Isotope Composition Between Xylem and Tissue Water in Populus Euphratica. Plant Cell Environ 39:1848–1857. doi: 10.1111/pce.12753

---

## Short Comment (SC1) · Comment on Barbeta et al. · 6 Mar 2019

Barbeta et al. argue that fractionation could have occurred upon uptake or within plants because they often observed that xylem water samples were lower in $\delta$2H than any of the potential sources they measured (rock water, stream water, fog, soils from 70-80 cm, and soils from 0-10 cm). They consider a few possible explanations (e.g., "separation between mobile and bound" and "compartmentalization between vessel water and other stem water pools"), but mostly they "argue that an isotopic fractionation in the unsaturated zone and/or within the plant tissues could underlie" their observations.

Of course evaporation causes fractionation in the unsaturated zone, which they show clear evidence of, but they are arguing that there may be an unexplained fractionation that occurs in stems or upon root uptake (similar to that which is sometimes observed in halophytes and xerophytes). Such an argument may be valid if a reasonably comprehensive set of potential explanations have been considered and rejected. However, they did not sample highly likely water sources, and thus there are very probable explanations for their results that were not considered.

In the introduction, the authors state "if H1 is true [i.e., that there is no fractionation upon root uptake], the $\delta 18O$ and $\delta 2H$ of xylem water should always lie within the range of values of all water sources." Thus, to test H1, all source waters should be sampled. Although sampling all source waters is an infeasible task, even highly likely sources were not measured (e.g., soils between 10 and 70 cm depth). Thus, the rejection of H1 is not a logical extension of this study's findings, and it is unclear why the authors focus on attributing their findings to fractionation upon uptake or during within-plant transport.

Isotopic fractionation during plant root uptake can be most accurately tested in controlled settings where the "true" value is predictable, not in ambient field conditions where there are many un-controlled complicating factors. Controlled experiments have shown that xylem water accurately reflects soil water isotope values (e.g., Newberry et al 2017); consequently, challenging those findings requires a robust, well-constrained experiment. In the present study, it is not clear that the observed differences between the sampled end members and xylem water samples are due to fractionation during uptake or within the plant, as opposed to numerous other likely explanations. Several of these are listed below.

1) No soil water samples were collected at depths where roots are often found (10 to 70 cm). Thus, the authors cannot exclude the possibility that the trees' apparent source waters occurred between their shallow (0-10 cm) and deep (>70 cm) samples. If these profiles were only affected by evaporation, then perhaps a profile comprising progressively enriched values towards the top could be expected. However, precipitation infiltrates and mixes heterogeneously with stored waters, creating heterogeneity and obscuring an evaporation profile (for an example that obviously expresses transport effects, see Figure 3 in Sprenger et al 2016). It should not be assumed that soils in intermediate depths (10-70 cm) have isotope values that are in between those of deeper and shallower soils (see Thomas et al., 2013). The un-sampled soil water domain could include winter precipitation that percolated downward into the rooting zone, after undergoing evaporative fractionation near the surface (yielding lower isotope values, due to the water's winter origins, with negative LC-excess, due to evaporation; e.g., see Dudley et al 2018), consistent with the xylem water values shown. My research (including two of the same species) shows that summer use of winter precipitation by plants is a reasonable expectation (Allen et al., 2019). It is reasonable to expect that zones between 10 cm and 70 cm contain roots, and contain winter precipitation with an evaporated signature. Thus, this constitutes a likely source that was entirely overlooked.

2) Laser spec analysis issues may compromise inferences. Of course the authors know that using a laser spec can yield uncertain xylem water measurements, and they made attempts to correct those data. However, given that the authors are challenging long-standing knowledge, it is essential to control for the potentially confounding effects of organics (not just "methanol and/or ethanol") in the laser spec analyses. Although the authors are more attentive to this issue than many, benchmarking a subset of the samples using IRMS would provide a more convincing data set.

3) Lateral heterogeneities create challenges for representative sampling. For the 0-10 cm depth, where soil water isotope signatures are most heterogeneous, there were relatively few samples collected. Three cores per plot is minimal. Goldsmith et al (2019; see Figure 7) show that dramatic mischaracterizations of the true variance among surface soil water isotope ratios should be expected when using small sample sizes. The authors cannot retroactively sample the soils, but they should recognize that their sampling probably underestimates the range of lateral variation. It could also be considered

that there are fine-scale variations in pore sizes that plants may differentially sample among (Stewart et al., 1999), but an auger cannot. Given that different pore sizes transport water at different rates, we should expect them to correspond with fine-scale variations in isotope values.

Given these limitations in the sampling and analysis (especially a lack of samples from 10-70 cm), it is unjustified to attribute the lack of finding an appropriate source to unexplained fractionation processes in stems or at the root-soil interface. A more defensible conclusion is that the specific sampling regime used here may not have captured the source waters that were actually used by the trees.

References Allen, S. T., Kirchner, J. W., Braun, S., Siegwolf, R. T., & Goldsmith, G. R. (2019). Seasonal origins of soil water used by trees. Hydrology and Earth System Sciences, 23(2), 1199-1210.

Dudley, B. D., Marttila, H., Graham, S. L., Evison, R., & Srinivasan, M. S. (2018). Water sources for woody shrubs on hillslopes: An investigation using isotopic and sapflow methods. Ecohydrology, 11(2), e1926.

Goldsmith, G. R., Allen, S. T., Braun, S., Engbersen, N., González‐Quijano, C. R., Kirchner, J. W., & Siegwolf, R. T. (2018). Spatial variation in throughfall, soil, and plant water isotopes in a temperate forest. Ecohydrology, e2059.

Newberry, S. L., Nelson, D. B., & Kahmen, A. (2017). Cryogenic vacuum artifacts do not affect plant water‐uptake studies using stable isotope analysis. Ecohydrology, 10(8), e1892.

Sprenger, M., Erhardt, M., Riedel, M., & Weiler, M. (2016). Historical tracking of nitrate in contrasting vineyards using water isotopes and nitrate depth profiles. Agriculture, Ecosystems & Environment, 222, 185-192.

Stewart, J., Moran, C. & Wood, J. Macropore sheath: quantification of plant root and soil macropore association. Plant and Soil (1999) 211: 59.

https://doi.org/10.1023/A:1004405422847

Thomas, E. M., H. Lin, C. J. Duffy, P. L. Sullivan, G. H. Holmes, S. L. Brantley, and L. Jin. 2013. Spatiotemporal Patterns of Water Stable Isotope Compositions at the Shale Hills Critical Zone Observatory: Linkages to Subsurface Hydrologic Processes. Vadose Zone J. 12. doi:10.2136/vzj2013.01.0029

————————————————————

---

## Author Comment (AC3) · 14 Mar 2019

**Response to Scott T Allen's short comment**

**Manuscript: hess-2018-631**

Comment on Barbeta et al.

Barbeta et al. argue that fractionation could have occurred upon uptake or within plants because they often observed that xylem water samples were lower in $\delta^2H$ than any of the potential sources they measured (rock water, stream water, fog, soils from 70-80 cm, and soils from 0-10 cm). They consider a few possible explanations (e.g., "separation between mobile and bound" and "compartmentalization between vessel water and other stem water pools"), but mostly they "argue that an isotopic fractionation in the unsaturated zone and/or within the plant tissues could underlie" their observations.

**General comment**

**We appreciate the interest that Dr. Allen raised on our study, and the time he dedicated to offer his point of view on the conclusion that we have drawn from it (the existence of an 2H/1H separation between plant xylem water and water sources). In short, Dr. Allen questions this conclusion based on three potential explanations (PEs) that, in his opinion, we have not considered: (PE1) non-monotonic variations of soil water isotope composition with depth, (PE2) laser spectral interferences with organics and (PE3) spatial heterogeneity of soil surface water isotope composition. Although these points raised by Dr. Allen were already addressed and discussed in our manuscript, it seems that extra clarifications are required. We added such clarifications in the revised manuscript, notably regarding PE3, and provided also a detailed point-by-point response below.**

Of course evaporation causes fractionation in the unsaturated zone, which they show clear evidence of, but they are arguing that there may be an unexplained fractionation that occurs in stems or upon root uptake (similar to that which is sometimes observed in halophytes and xerophytes). Such an argument may be valid if a reasonably comprehensive set of potential explanations have been considered and rejected. However, they did not sample highly likely water sources, and thus there are very probable explanations for their results that were not considered.

In the introduction, the authors state "if H1 is true [i.e., that there is no fractionation upon root uptake], the $\delta^{18}O$ and $\delta^2H$ of xylem water should always lie within the range of values of all water sources." Thus, to test H1, all source waters should be sampled. Although sampling all source waters is an infeasible task, even highly likely sources were not measured (e.g., soils between 10 and 70 cm depth). Thus, the rejection of H1 is not a logical extension of this study's findings, and it is unclear why the authors focus on attributing their findings to fractionation upon uptake or during within-plant transport.

Isotopic fractionation during plant root uptake can be most accurately tested in controlled settings where the "true" value is predictable, not in ambient field conditions where there are many un-controlled complicating factors. Controlled experiments have shown that xylem water accurately reflects soil water isotope values (e.g., Newberry et al 2017); consequently, challenging those findings requires a robust, well-constrained experiment. In the present study, it is not clear that the observed differences between the sampled end members and xylem water samples are due to fractionation during uptake or within the plant, as opposed to numerous other likely explanations. Several of these are listed below.

**The rejection of H1 is not a logical extension of our study, as stated by the Dr. Allen. A careful read of our manuscript should not give the reader such impression. Our results are rather embedded in a growing series of papers reporting isotopic offsets in the field (Brooks *et al.*, 2010; Geris *et al.*, 2015; Evaristo *et al.*, 2016, 2017; Wang *et al.*, 2017; De Deurwaerder *et al.*, 2018), but importantly also in controlled experiments (Vargas *et al.*, 2017). Still, further studies are needed to formally reject the mentioned H1.**

**We agree with Dr. Allen that the rejection of H1 requires a controlled test where the true isotopic value of xylem water is predictable. However, it is noteworthy that the list of controlled experiments actually confirming H1 under controlled experiments is rather short. Dr. Allen cites one single recent paper (Newberry *et al.*, 2017) that did not show isotopic offsets between soil and xylem water. However, two also recent studies showed opposite results (*i.e.* rejecting H1) (explicitly in Vargas *et al.*, 2017; Orlowski *et al.*, 2018). These are not even novel findings as isotopic fractionation occurring during root water uptake had been suggested more than three decades ago (Allison *et al.*, 1983). Importantly, we have since conducted a similar controlled experiment on *F. sylvatica* saplings in which we confirm the occurrence of a 2H/1H fractionation between stem and soil water of the same magnitude as reported here (Barbeta et al. in preparation). We decided to conduct this controlled experiment after a thorough evaluation of the "numerous" likely explanations for the field results reported in the current manuscript. Below, we explain in more details why the explanations proposed by Dr. Allen were found not plausible, and also clarified these explanations in the main text.**

1) No soil water samples were collected at depths where roots are often found (10 to 70 cm). Thus, the authors cannot exclude the possibility that the trees' apparent source waters occurred between their shallow (0-10 cm) and deep (>70 cm) samples. If these profiles were only affected by evaporation, then perhaps a profile comprising progressively enriched values towards the top could be expected. However, precipitation infiltrates and mixes heterogeneously with stored waters, creating heterogeneity and obscuring an evaporation profile (for an example that obviously expresses transport effects, see Figure 3 in Sprenger et al 2016). It should not be assumed that soils in intermediate depths (10-70 cm) have isotope values that are in between those of deeper and shallower soils (see Thomas et al., 2013). The un-sampled soil water domain could include winter precipitation that percolated downward into the rooting zone, after undergoing evaporative fractionation near the surface (yielding lower isotope values, due to the water's winter origins, with negative LC-excess, due to evaporation; e.g., see Dudley et al 2018), consistent with the xylem water values shown. My research (including two of the same species) shows that summer use of winter precipitation by plants is a reasonable expectation (Allen et al., 2019). It is reasonable to expect that zones between 10 cm and 70 cm contain roots, and contain winter precipitation with an evaporated signature. Thus, this constitutes a likely source that was entirely overlooked.

**Our sampling strategy was designed to capture as much as possible the spatio-temporal variability in soil water isotopes, while keeping the analytical cost within reason. With the aim of optimizing the sampling effort (and sampling processing in the lab) we purposely restricted our sampling of water sources to top soil layers exposed to evaporation (0-10 cm) and deep soil layers (below 60cm) only affected by infiltration and mixing processes, and thus expected to display less variability over the season. Indeed, based on soil texture and climate, we did not expect soil evaporation to affect these deep soil layers at our field site. This was confirmed by a detailed soil isotopic profile collected at the end of the summer in September 2018 (Figure SC1a below).**

[Figure]

**Fig. SC1a. δ1. Mean (±SE, *N* = X locations per depth) soil water isotopic composition ($\delta^{18}O$ and $\delta^{2}H$) at different depths. Different letters indicate significant differences among depths (*P*<0.05).**

**From this figure we see that there is no significant difference in the $\delta^{18}O$ and $\delta^{2}H$ of soil water among different depths below 20 cm, while the $\delta^{18}O$ and $\delta^{2}H$ of the upper layers are more enriched (not more depleted). We acknowledge that this isotopic profile could change over the course of the season, for instance following a rain event. Summer rain would deplete the topsoil layers but never to values more negative than winter precipitation, and would also add noise to the soil water line regression. In our revised version we have included the statistics of the soil water line regressions for the different sampling campaigns, following the comments of reviewer Juan Pedro Ferrio. In the response to Dr. Ferrio's comments, we included the modifications done in the text to acknowledge that the absence of such an isotopic profile caused by evaporation can lead to uncertainties when applying the SW-excess approach. In cases where the regression of the soil water line was not significant, the calculated isotopic offsets may be less meaningful. However, this only happened in a few cases (Table S3 in the revised manuscript). In addition, the soil water lines were calculated at the plot-scale and for every single date (Table S3), and the fit of these lines did not affect the estimated SW-excess.**

Dr. Allen also pointed to an overlooked water source, namely winter precipitation stored in this middle soil layer. A feature of our study site is the varying soil texture in depth. In the allegedly overlooked soil layer (10 to 70 cm depth), the soil texture is coarse sand (Table S1) and the rock fraction is practically zero. On the other hand, the deeper soil layer that we targeted with our sampling strategy is a sandy loam, with higher water retention capacity (Figure SC1b below). Volumetric soil water content data from 2018 suggests that the water storage between 10 and 70 cm layers is minimal during summer, whereas the deeper soil layer holds more moisture all throughout the growing season (Figure SC1b). This is probably caused by higher infiltration rates of the coarse sand horizon, that we did not systematically sample. Although in terms of soil water potential (and thus extractable soil water) these different soil layers should not differ too much, it is very unlikely that the sandy layer would be able to hold winter precipitation until summer. On the other hand, what we sampled as representative for the isotopically unenriched part of the soil (>70cm) is also replenished during winter, which is very rainy in the area. Deep soil only starts drying out in summer, late summer precipitation does not infiltrate into the deeper soil layers, where soil water content only recovers after the first autumn storms (Fig. SC1b). Although we do not have such depth-resolved information for 2017, the GWC data presented in this manuscript illustrates a similar seasonal pattern (Fig. 1 of the manuscript). Importantly, winter rain is depleted in both $\delta^{18}O$ and $\delta^{2}H$ but in our study, xylem water $\delta^{18}O$ was always in the range of soil water $\delta^{18}O$. To sum up, we did not 'entirely overlooked' winter precipitation, as this is very likely the main source of the deep soil layers that we systematically sampled throughout the growing season.

[Figure]

**Fig. SC1b. Volumetric water content (Hum) at different depths in one of our study sites during the growing season of 2018. Horizon A (10 and 20cm), horizon B (30, 40 and 60cm), horizon C (100cm).**

2) Laser spec analysis issues may compromise inferences. Of course the authors know that using a laser spec can yield uncertain xylem water measurements, and they made attempts to correct those data. However, given that the authors are challenging long-standing knowledge, it is essential to control for the potentially confounding effects of organics (not just "methanol and/or ethanol") in the laser spec

analyses. Although the authors are more attentive to this issue than many, benchmarking a subset of the samples using IRMS would provide a more convincing data set.

**Indeed, infrared isotope spectrometers (IRIS) are known to be sensitive to organic volatiles that also absorb light in the mid infrared range explored by these analyzers. IRMS are also sensitive to organic compounds but to an extent that is only proportional to mass contribution of these compounds (Martín-Gómez *et al.*, 2015). As we report in the manuscript, we developed a correction specific for our instrument following previous studies (Schultz *et al.*, 2011; Brian Leen *et al.*, 2012). We found that the corrections applied to the data could not possibly explain the observed SW-excess. In addition, as briefly mentioned in the manuscript, soil-xylem $\delta^2$H offsets of similar magnitude have been reported by other investigators using both IRIS and IRMS, even in controlled settings. Here is a non-exhaustive list of them:**

**Table SC1a. List of studies showing soil-xylem isotopic offsets comparable to those found in the present study.**

| Study | Analytical method | Experiment type, study species |
|---|---|---|
| Geris *et al.*, (2015) | IRMS | Field, *Pinus sylvestris* |
| Vargas *et al.*, (2017) | IRMS | Glasshouse, *Persea americana* |
| Evaristo *et al.*, (2017) | IRMS | Botanical garden, many species |
| Wang *et al.*, (2017) | IRMS | Field, deciduous shrubs and perennial herb |
| De Deurwaerder *et al.*, (2018) | IRIS (Picarro with MCM) | Field, rainforest tree species |
| Brooks *et al.*, (2010) | IRIS (LGR) & IRMS | Field, *Pseudotsuga menziensii* |
| Evaristo *et al.*, (2016) | IRIS (Picarro) & IRMS | Field, *Swietenia macrophylla* |

3) Lateral heterogeneities create challenges for representative sampling. For the 0-10 cm depth, where soil water isotope signatures are most heterogeneous, there were relatively few samples collected. Three cores per plot is minimal. Goldsmith et al (2019; see Figure 7) show that dramatic mischaracterizations of the true variance among surface soil water isotope ratios should be expected when using small sample sizes. The authors cannot retroactively sample the soils, but they should recognize that their sampling probably underestimates the range of lateral variation. It could also be considered that there are fine-scale variations in pore sizes that plants may differentially sample among (Stewart et al., 1999), but an auger cannot.

Given that different pore sizes transport water at different rates, we should expect them to correspond with fine-scale variations in isotope values. Given these limitations in the sampling and analysis (especially a lack of samples from 10-70 cm), it is unjustified to attribute the lack of finding an appropriate source to unexplained fractionation processes in stems or at the root-soil interface. A more defensible conclusion is that the specific sampling regime used here may not have captured the source waters that were actually used by the trees.

**We agree with Dr. Allen that the surface spatial heterogeneity in soil water isotopes can be large. Dr. Allen finds that three cores per plot is minimal. Maybe it was not entirely clear in our Methods section but these plots were relatively small (maximum distance between the two most distant trees was 15 m), and all trees within the plot had a soil core below their canopies. Of course, our sampling design**

cannot ensure that all the variability in surface soil water isotopes is captured. Horizontal heterogeneity in soil water isotopic composition would certainly generate noise on the SW-excess determination but cannot explain the isotopic offsets between soil and xylem water observed over the entire growing season (over a wide range of environmental conditions) and for all the studied trees (representing differences in size, species, plot, and thus very likely also in rooting depth and lateral root spread).

In the last paragraph of his comment, Dr. Allen points out again to limitations of our sampling design. On the contrary, a strong point of our study is the detailed characterization of the temporal dynamics (bi-weekly sampling campaigns sustained for a whole growing season) of soil water, together with rock moisture, groundwater and stream and fog water. Despite the issues related with cryogenic extraction, which we already consider and discuss in the text, cryogenically extracted water is still a good proxy for soil water isotopes (Newberry *et al.*, 2017). This is especially true when compared with lysimeter-extracted water that subsample soil mobile water, and that is not representative of plant-accessible water that can be held at down to -1500 kPa or more (Slatyer, 1957). A promising avenue for advancing our understanding is the deployment of systems allowing continuous measurements of soil water isotope vapor profiles *in situ* (Oerter *et al.*, 2019), which would provide a different perspective on the spatio-temporal patterns of water isotopes in the soil-root interface.

References (cited by the Dr. Allen)

Allen, S. T., Kirchner, J. W., Braun, S., Siegwolf, R. T., & Goldsmith, G. R. (2019). Seasonal origins of soil water used by trees. Hydrology and Earth System Sciences, 23(2), 1199-1210.

Dudley, B. D., Marttila, H., Graham, S. L., Evison, R., & Srinivasan, M. S. (2018). Water sources for woody shrubs on hillslopes: An investigation using isotopic and sapflow methods. Ecohydrology, 11(2), e1926.

Goldsmith, G. R., Allen, S. T., Braun, S., Engbersen, N., González Quijano, C. R., Kirchner, J. W., & Siegwolf, R. T. (2018). Spatial variation in throughfall, soil, and plant water isotopes in a temperate forest. Ecohydrology, e2059.

Newberry, S. L., Nelson, D. B., & Kahmen, A. (2017). Cryogenic vacuum artifacts do not affect plant waterâA˘ Ruptake studies using stable isotope analysis. Ecohydrology, 10(8), e1892. Sprenger, M., Erhardt, M., Riedel, M., & Weiler, M. (2016). Historical tracking of nitrate in contrasting vineyards using water isotopes and nitrate depth profiles. Agriculture, Ecosystems & Environment, 222, 185-192.

Stewart, J., Moran, C. & Wood, J. Macropore sheath: quantification of plant root and soil macropore association. Plant and Soil (1999) 211: 59. https://doi.org/10.1023/A:1004405422847

Thomas, E. M., H. Lin, C. J. Duffy, P. L. Sullivan, G. H. Holmes, S. L. Brantley, and L. Jin. 2013. Spatiotemporal Patterns of Water Stable Isotope Compositions at the Shale Hills Critical Zone Observatory: Linkages to Subsurface Hydrologic Processes. Vadose Zone J. 12. doi:10.2136/vzj2013.01.0029.

**References (authors' response)**

**Allison GB, Barnes CJ, Hugues MW, Leaney FWJ**. **1983**. Effect of Climate and Vegetation on Oxygen-18 and Deuterium Profiles in Soils. *Isotope hydrology, 1983 : proceedings of an International Symposium on Isotope Hydrology in Water Resources Development*: 105–123.

**Brian Leen J, Berman ESF, Liebson L, Gupta M**. **2012**. Spectral contaminant identifier for off-axis integrated cavity output spectroscopy measurements of liquid water isotopes. *Review of Scientific Instruments* **83**(4).

**Brooks JR, Barnard HR, Coulombe R, McDonnell JJ, Renée Brooks J, Barnard HR, Coulombe R, McDonnell JJ**. **2010**. Ecohydrologic separation of water between trees and streams in a Mediterranean climate. *Nature Geoscience* **3**: 100–104.

**De Deurwaerder H, Hervé-Fernández P, Stahl C, Burban B, Petronelli P, Hoffman B, Bonal D, Boeckx P, Verbeeck H**. **2018**. Liana and tree below-ground water competition—evidence for water resource partitioning during the dry season. *Tree Physiology* 38, 1071–1083.

**Evaristo J, McDonnell JJ, Clemens J**. **2017**. Plant source water apportionment using stable isotopes: A comparison of simple linear, two-compartment mixing model approaches. *Hydrological Processes* **31**: 3750–3758.

**Evaristo J, McDonnell JJ, Scholl MA, Bruijnzeel LA, Chun KP**. **2016**. Insights into plant water uptake from xylem-water isotope measurements in two tropical catchments with contrasting moisture conditions. *Hydrological Processes* **30**: 3210–3227.

**Geris J, Tetzlaff D, McDonnell J, Anderson J, Paton G, Soulsby C**. **2015**. Ecohydrological separation in wet, low energy northern environments? A preliminary assessment using different soil water extraction techniques. *Hydrological Processes* **29(25)**, 5139-5152.

**Martín-Gómez P, Barbeta A, Voltas J, Peñuelas J, Dennis K, Palacio S, Dawson TE, Ferrio JP**. **2015**. Isotope-ratio infrared spectroscopy: A reliable tool for the investigation of plant-water sources? *New Phytologist*. **207**(3): 914-27.

**Newberry SL, Nelson DB, Kahmen A**. **2017**. Cryogenic vacuum artifacts do not affect plant water-uptake studies using stable isotope analysis. *Ecohydrology* **10**(8): e1892.

**Oerter EJ, Siebert G, Bowling DR, Bowen G**. **2019**. Soil water vapor isotopes identify missing water source for streamside trees. *Ecohydrology*: e2083.

**Orlowski N, Winkler A, McDonnell JJ, Breuer L**. **2018**. A simple greenhouse experiment to explore the effect of cryogenic water extraction for tracing plant source water. *Journal of Professions and Organization* **11(5)**: e1967.

**Schultz NM, Griffis TJ, Lee X, Baker JM**. **2011**. Identification and correction of spectral contamination in 2H/ 1H and 18O/ 16O measured in leaf, stem, and soil water. *Rapid Communications in Mass Spectrometry* **25**: 3360–3368.

**Slatyer RO**. **1957**. The significance of the permanent wilting percentage in studies of plant and soil water relations. *The Botanical Review* **XXIII**: 585–636.

**Vargas AI, Schaffer B, Yuhong L, Sternberg L da SL**. **2017**. Testing plant use of mobile vs immobile soil water sources using stable isotope experiments. *New Phytologist* **215**: 582–594.

**Wang J, Fu B, Lu N, Zhang L**. **2017**. Seasonal variation in water uptake patterns of three plant species based on stable isotopes in the semi-arid Loess Plateau. *Science of the Total Environment* **609**: 27–37.

---

## Short Comment (SC2) · 22 Mar 2019

In my previous comment on Barbeta et al, I suggested that the mismatch between xylem waters and their measured potential source waters may be due to root water uptake from soils that were not sampled. Soils between depths of 10 and ~70 cm samples were not sampled, which is a large range from which roots often take up water. However, the authors' response states that such an explanation was "not found plausible" and offer the following explanation:

"Our sampling strategy was designed to capture as much as possible the spatio-temporal variability in soil water isotopes, while keeping the analytical cost within reason. With the aim of optimizing the sampling effort (and sampling processing in the lab) we purposely restricted our sampling of water sources to top soil layers exposed to evaporation (0-10 cm) and deep soil layers (below 60cm) only affected by infiltration and mixing processes, and thus expected to display less variability over the season." Indeed, based on soil texture and climate, we did not expect soil evaporation to affect these deep soil layers at our field site. This was confirmed by a detailed soil isotopic profile collected at the end of the summer in September 2018 (Figure SC1a below).

From this figure we see that there is no significant difference in the δ18O and δ 2H of soil water among different depths below 20 cm, while the δ 18O and δ 2H of the upper layers are more enriched (not more depleted). We acknowledge that this isotopic profile could change over the course of the season, for instance following a rain event. Summer rain would deplete the topsoil layers but never to values more negative than winter precipitation, and would also add noise to the soil water line regression. ... "

I understand that the cost of sampling limits how the sampling can be conducted; however, this also limits the potential inferences. I do not understand the justification for the authors' assumptions about the unsampled depths. While isotopic variations in the shallowest and deep soils may be a product of different processes, this does not imply that they are bounds for the full range of isotope values. I believe that the new figure (SC1a), showing a profile from a single time that they selected, demonstrates the possibility of a non-monotonic profile where intermediate soil depths contain isotope values that are not bounded by shallower and deeper isotope values. The intermediate soil depths (e.g., 20-50 cm) contain $\delta^2H$ and $\delta^{18}O$ values that are lower than those of the shallowest or deepest soils. The "no significant difference" may arise because those intermediate depths are highly variable and contain a wide range of isotope ratios (see the larger SE values at 20-35 cm). While they display SEs, which are measures of confidence in means, those SE values are smaller than the full range of values (which may be more relevant for supporting the argument that the missing source value could not exist in intermediate soils depths). Regardless, this one snapshot into a profile suggests that intermediate values can have lower deuterium values than those seen in the shallowest or deepest measurements.

Upper soils are also affected by transport processes. Summer rains do not "deplete the topsoil layers", but instead mix with shallow soil water, yielding a new (probably lower) $\delta^{18}O$ or $\delta^2H$ value, and / or displace that (former) surface water, causing it to percolate downward. Thus downward percolating water may have previously been in the shallowest soils (and underwent strong evaporative fractionation so it had low LC-excess), and may be sourced from previous precipitation events (e.g., potentially by precipitation inputs that had especially low $\delta^{18}O$ and $\delta^2H$ prior to evaporation effects).

See Figure 6 in Barbecot et al 2018, which shows that LC-excess values in ~20-30 cm depth are lower than those in the shallowest soils, and that these "evaporated" signals are associated with low $\delta^{18}O$ values. Note that one of the sites used in this study is from a sandy soil in France, and that sandy soil preserves prior seasons' precipitation signals.

See Figure 4 in in Oerter et al 2019, which shows that a) the most negative $\delta^2H$ values often occurred in intermediate depths, b) lower LC-excess values can vary non-monotonically (suggesting downward

transport of previously evaporated waters). They state, "*LC-excess* values were relatively high (near 0‰, Figure [4]c) in the upper 10- to 15-cm soil depth from April through mid-June, which indicates that the higher $\delta^2 H_{liq}$ and $\delta^{18}O_{liq}$ values at the shallow depths during this time were due to spring season precipitation with higher $\delta^2 H$ and $\delta^{18}O$ values, rather than being caused by evaporative enrichment in shallow soil water (Figure [4]c). Deep soil *LC-excess* values below 20 cm from April through the end of June were approximately −30‰, with relatively little depth variability. These deep soil *LC-excess* values bearing an evaporative signal are likely derived from winter snowmelt that was partially evaporated or sublimated prior to infiltration." Perhaps most importantly, Oerter et al use these data to argue that previously described mismatches between xylem water and soil waters may be due to previous researchers' limited sampling of soil waters.

To be perfectly clear, I am not saying that manuscript in discussion does not show interesting data. Nor am I saying that we should rule out the possibility that fractionation occurs upon uptake. The authors suggest that they have a forthcoming paper that demonstrates fractionation upon uptake, which would be a very useful thing to demonstrate. Nonetheless, I remain uncertain why it is implausible that the missing source (in this study) might be soil water from depths between 10 and 70 cm, where isotope ratios can be heterogeneous and variable.

Oerter, EJ, Siebert, G, Bowling, DR, Bowen, G. Soil water vapour isotopes identify missing water source for streamside trees. *Ecohydrology*. 2019;e2083. https://doi.org/10.1002/eco.2083

Barbecot F, Guillon S, Pili E, Larocque M, Gibert-Brunet E, et al.. Using Water Stable Isotopes in the Unsaturated Zone to Quantify Recharge in Two Contrasted Infiltration Regimes. *Vadose Zone Journal*, 2018, 17 (1), pp.1-13.

---

## Author Comment (AC4) · 26 Mar 2019

**Response to SC2 by Dr Scott Allen**

In this second comment, Dr. Allen further developed his arguments on the possibility of a missing water source that could explain our observation of a persistent hydrogen isotope composition ($\delta^2$H) offset between plant xylem water and all considered water sources. Although the mechanisms brought about by Dr. Allen are sound, they do not give a quantitative explanation of why the offset would occur for $\delta^2$H only, and not $\delta^{18}$O. Dr. Allen quoted only a fragment of our response, but we had already explained why a non-monotonic soil isotopic profile should impact both $\delta^2$H and $\delta^{18}$O (although not necessarily in exactly the same way). Nonetheless, to fully address Dr. Allen's comments, we have reinforced the idea in our manuscript that a more detailed sampling design (i.e. with more soil water samples from intermediate depths) would have helped capture better the SW-excess and raised any doubt on the existence and persistence of these "unexplained hydrogen isotope offsets" (to quote our own title).

Dr. Allen also brought up a study that explained isotopic mismatches by isotopic differences between the soil water pools that are accessed by plants, using a recently developed technique (Oerter *et al.*, 2019). This response is not the place to comment on this other study. Instead, we would like to mention that the same authors have also published another study, using the same technique, where they conclude to have "found some support for H isotope discrimination effects during water uptake by *Quercus gambelii*" (Oerter & Bowen, 2019). This is in full agreement with the results and conclusions of our own study. The reason why studies, even from the same author, have contrasting results is because the mechanisms leading to the "unexplained hydrogen isotope offsets" have not been identified yet and may vary between soil types, plant species and climate conditions. However, we feel it is rather dishonest to systematically ignore studies that claim that H isotope discrimination effects during plant water uptake may occur. Instead we offer the reader an array of explanations for our results, and remain rather cautious when it comes to identifying the exact mechanisms.

**A point-by-point response is provided below.**

Response to Barbeta et al.

In my previous comment on Barbeta et al, I suggested that the mismatch between xylem waters and their measured potential source waters may be due to root water uptake from soils that were not sampled. Soils between depths of 10 and ~70 cm samples were not sampled, which is a large range from which roots often take up water. However, the authors' response states that such an explanation was "not found plausible" and offer the following explanation:

*"Our sampling strategy was designed to capture as much as possible the spatio-temporal variability in soil water isotopes, while keeping the analytical cost within reason. With the aim of optimizing the sampling effort (and sampling processing in the lab) we purposely restricted our sampling of water sources to top soil layers exposed to evaporation (0-10 cm) and deep soil layers (below 60cm) only affected by infiltration and mixing processes, and thus expected to display less variability over the season." Indeed, based on soil texture and climate, we did not expect soil evaporation to affect these deep soil layers at our field site. This was confirmed by a detailed soil isotopic profile collected at the end of the summer in September 2018 (Figure SC1a below).*

*From this figure we see that there is no significant difference in the $\delta^{18}$O and $\delta^2$H of soil water among different depths below 20 cm, while the $\delta^{18}$O and $\delta^2$H of the upper layers are more enriched (not more*

*depleted). We acknowledge that this isotopic profile could change over the course of the season, for instance following a rain event. Summer rain would deplete the topsoil layers but never to values more negative than winter precipitation, and would also add noise to the soil water line regression. … "*

I understand that the cost of sampling limits how the sampling can be conducted; however, this also limits the potential inferences. I do not understand the justification for the authors' assumptions about the unsampled depths. While isotopic variations in the shallowest and deep soils may be a product of different processes, this does not imply that they are bounds for the full range of isotope values. I believe that the new figure (SC1a), showing a profile from a single time that they selected, demonstrates the possibility of a non-monotonic profile where intermediate soil depths contain isotope values that are not bounded by shallower and deeper isotope values. The intermediate soil depths (e.g., 20-50 cm) contain $\delta^2$H and $\delta^{18}$O values that are lower than those of the shallowest or deepest soils. The "no significant difference" may arise because those intermediate depths are highly variable and contain a wide range of isotope ratios (see the larger SE values at 20-35 cm). While they display SEs, which are measures of confidence in means, those SE values are smaller than the full range of values (which may be more relevant for supporting the argument that the missing source value could not exist in intermediate soils depths). Regardless, this one snapshot into a profile suggests that intermediate values can have lower deuterium values than those seen in the shallowest or deepest measurements.

**The reason why we presented a detailed soil water isotopic profile in our previous response was not to refute the idea that non-monotonic soil water isotopic profiles could have occurred under certain circumstances. We agree that this isotopic profile does show some soil water isotope depletion in intermediate depths (20-35cm), although not significant. However, as we already mentioned in our response to the first comment, such a non-monotonic profile would imply that we should see also a depleted xylem water $\delta^{18}$O, which was not the case. In addition, such non-monotonic isotopic profiles would need to be sustained over time and space, because the isotopic offset was observed in all campaigns and sampled trees. Xylem $\delta^2$H was persistently more depleted than soil water $\delta^2$H, that is, following rain events and subsequent mixing processes, under relatively dry conditions, both at the beginning of the season and right before autumn leaf shedding, representing a wide range of soil moisture (and isotopic) conditions as well as probably different prevalent depths of water uptake.**

Upper soils are also affected by transport processes. Summer rains do not "deplete the topsoil layers", but instead mix with shallow soil water, yielding a new (probably lower) $\delta^{18}$O and $\delta^2$H value, and/or displace that (former) surface water, causing it to percolate downward. Thus downward percolating water may have previously been in the shallowest soils (and underwent strong evaporative fractionation so it had low LC-excess), and may be sourced from previous precipitation events (e.g., potentially by precipitation inputs that had especially low $\delta^{18}$O and $\delta^2$H prior to evaporation effects).

**If summer rain mixes with shallow soil water, it should "deplete the top soil layers" (via mixing) because it is expected to be depleted compared to an initial surface soil water that has undergone soil evaporation. Dr. Allen argues that alternatively winter rain may have been displaced to intermediate depths by subsequent spring rain events, without mixing, and stayed there throughout the growing season. As explained in our answer, the intermediate soil layers are very sandy and cannot hold much water. It is thus much more likely that winter precipitation would remain in the deep soil layers that we sampled.**

See Figure 6 in Barbecot et al 2018, which shows that LC-excess values in ~20-30 cm depth are lower than those in the shallowest soils, and that these "evaporated" signals are associated with low $\delta^{18}O$ values. Note that one of the sites used in this study is from a sandy soil in France, and that sandy soil preserves prior seasons' precipitation signals.

**It is not clear to us what is the link between the data shown by Barbecot et al. and our results. First, the data is from a single sampling date, which does not imply a persistent pattern over the growing season, as we observed. Second, Dr. Allen highlights that the LC-excess was more negative in the 20-30 cm that in shallower layers (for that particular date), but as he correctly mentions as well, those negative LC-excess are associated with low $\delta^{18}O$ values. In Fig. 3 of the same paper, it is clear that the modeled $\delta^{18}O$ and $\delta^2H$ of infiltrating water are proportional, i.e., they follow the same temporal pattern. Again, we can assume that an intermediate soil layer is isotopically depleted compared to shallower and deeper layers, but as shown by Barbecot et al (2018), this depletion should be occur for both $\delta^{18}O$ and $\delta^2H$.**

See Figure 4 in in Oerter et al 2019, which shows that a) the most negative $\delta^2H$ values often occurred in intermediate depths, b) lower LC-excess values can vary non-monotonically (suggesting downward transport of previously evaporated waters). They state, "LC-excess values were relatively high (near 0‰, Figure 4c) in the upper 10- to 15-cm soil depth from April through mid-June, which indicates that the higher $\delta^2H$ liq and $\delta^{18}O$ liq values at the shallow depths during this time were due to spring season precipitation with higher $\delta^2H$ and $\delta^{18}O$ values, rather than being caused by evaporative enrichment in shallow soil water (Figure 4c). Deep soil LC-excess values below 20 cm from April through the end of June were approximately −30‰, with relatively little depth variability. These deep soil LC-excess values bearing an evaporative signal are likely derived from winter snowmelt that was partially evaporated or sublimated prior to infiltration." Perhaps most importantly, Oerter et al use these data to argue that previously described mismatches between xylem water and soil waters may be due to previous researchers' limited sampling of soil waters.

**We appreciate the review of previous literature brought up by Dr. Allen. We have already read Oerter et al. (2019), they present an interesting dataset of vapor and liquid water isotopes. In their Fig. 4, the clearest pattern we can see is a rather gradual depletion of soil water $\delta^{18}O$ and $\delta^2H$ with depth, i.e. similar to what we found at our field site. It is true that the most depleted layer is not always the deepest, but the one ranging from 60 to 80 cm. We agree that this could suggest percolation of evaporated water, but this could also suggest different infiltration depths of rainwater from events with distinct isotopic composition. We leave these interpretations for those who know well that site and analyzed the data.**

**As we interpreted it, Oerter et al. (2019) report that soil water vapor probes were able to solve the isotopic mismatches of previous studies in that same site, not other ones. All along the paper, we did not find the statement mentioned by Dr. Allen that "described mismatches between xylem water and soil waters may be due to previous researchers' limited sampling of soil waters". Rather, they state that "Part of the problem in finding the missing water source supplying plant transpiration in some ecosystems, may be an inability to adequately quantify the isotopic composition of potential soil water pools". Specifically, they are referring to different soil water pools that could be accessed or not by roots and that may differ in their isotopic composition. This is already discussed in our manuscript (2[nd] paragraph of the Discussion).**

**Importantly, a study published during the discussion of our manuscript, also led by Erik Oerter (Oerter & Bowen, 2019) and using the very same technique (soil and root vapor probes) found very similar isotopic mismatches between soil and plant water isotopes to the ones reported by us. They conclude that while further investigations are needed (we agree), there is some evidence for isotopic fractionation during root water uptake.**

To be perfectly clear, I am not saying that manuscript in discussion does not show interesting data. Nor am I saying that we should rule out the possibility that fractionation occurs upon uptake. The authors suggest that they have a forthcoming paper that demonstrates fractionation upon uptake, which would be a very useful thing to demonstrate. Nonetheless, I remain uncertain why it is implausible that the missing source (in this study) might be soil water from depths between 10 and 70 cm, where isotope ratios can be heterogeneous and variable.

**We appreciate the interest of Dr. Allen in our work. We hope that we have argued satisfactorily our points of view, so the potential reader may create her/his own. We have now amended our manuscript to acknowledge the possibility of an effect of our sampling strategy on our results. We have done so by including all the arguments that we consider relevant, either provided by Dr. Allen, or by our own responses. Please see the revised version of the manuscript in the next step of the peer-review process. Finally, it is worth saying that our goal is not to demonstrate the existence of isotopic discrimination during root water uptake, but to test specific hypothesis formulated based on previous research, and to do so with data obtained through the most rigorous possible experimental design.**

References cited by Dr. Allen

Oerter, EJ, Siebert, G, Bowling, DR, Bowen, G. Soil water vapour isotopes identify missing water source for streamside trees. Ecohydrology. 2019;e2083. https://doi.org/10.1002/eco.2083

Barbecot F, Guillon S, Pili E, Larocque M, Gibert-Brunet E, et al.. Using Water Stable Isotopes in the Unsaturated Zone to Quantify Recharge in Two Contrasted Infiltration Regimes. Vadose Zone Journal, 2018, 17 (1), pp.1-13.

**References**

**Oerter EJ, Bowen GJ**. **2019**. Spatiotemporal heterogeneity in soil water stable isotopic composition and its ecohydrologic implications in semi-arid ecosystems. *Hydrological Processes*: 0–2.

**Oerter EJ, Siebert G, Bowling DR, Bowen G**. **2019**. Soil water vapor isotopes identify missing water source for streamside trees. *Ecohydrology*: e2083.